# Auditory information enhances post-sensory visual evidence during rapid multisensory decision-making

Léon Franzen [1,2,5] ✉, Ioannis Delis [3,5], Gabriela De Sousa[1,5], Christoph Kayser [4] &
Marios G. Philiastides [1] ✉

Despite recent progress in understanding multisensory decision-making, a conclusive mechanistic account of how the brain translates the relevant evidence into a decision is lacking. Specifically, it remains unclear whether perceptual improvements during rapid multisensory decisions are best explained by sensory (i.e., 'Early') processing benefits or post-sensory (i.e., 'Late') changes in decision dynamics. Here, we employ a well-established visual object categorisation task in which early sensory and post-sensory decision evidence can be dissociated using multivariate pattern analysis of the electroencephalogram (EEG). We capitalize on these distinct neural components to identify when and how complementary auditory information influences the encoding of decision-relevant visual evidence in a multisensory context. We show that it is primarily the post-sensory, rather than the early sensory, EEG component amplitudes that are being amplified during rapid audiovisual decision-making. Using a neurally informed drift diffusion model we demonstrate that a multisensory behavioral improvement in accuracy arises from an enhanced quality of the relevant decision evidence, as captured by the post-sensory EEG component, consistent with the emergence of multisensory evidence in higher-order brain areas.

[1] Institute of Neuroscience and Psychology, University of Glasgow, Glasgow, UK. [2] Centre for Sensory Studies, Concordia University, Montréal, Canada. [3] School of Biomedical Sciences, University of Leeds, Leeds, UK. [4] Department for Cognitive Neuroscience & Cognitive Interaction Technology – Center of Excellence, Bielefeld University, Bielefeld, Germany. [5] These authors contributed equally: Léon Franzen, Ioannis Delis, Gabriela De Sousa. ✉email: leon.franzen@mail.com; marios.philiastides@glasgow.ac.uk

In everyday life, we often encounter situations that demand rapid decisions based on ambiguous sensory information. Consolidating the available evidence requires processing information presented in more than one sensory modality and exploiting this for multisensory decision-making[1–4]. For example, the decision to cross a street on a foggy morning will be based on a combination of visual evidence about hazy objects in your field of view and muffled sounds from various sources.

The presence of complimentary audiovisual (AV) information can improve our ability to make perceptual decisions, when compared to visual (V) information alone[5–8]. While recent studies have provided a detailed picture of the emergence of different types of uni- and multisensory representations in the brain[4,9–11], these studies have not provided a conclusive mechanistic account of how the brain encodes and ultimately translates the relevant sensory evidence into a decision[2]. Specifically, it remains unclear whether the perceptual improvements of multisensory decision-making are best explained by a benefit in the early encoding of sensory information, changes in the efficiency of post-sensory processes, such as the accumulation of evidence, or changes in the required amount of accumulated evidence before committing to a choice.

These questions can be addressed within the general framework of sequential sampling models, such as the drift diffusion model (DDM), which posit that decisions are formed by a stochastic accumulation of evidence over time[12–16]. The DDM decomposes behavioral data into internal processes that reflect the rate of evidence accumulation (drift rate), the amount of evidence required to make a decision (starting point and decision boundaries corresponding to the different decision alternatives), and latencies induced by early stimulus encoding and response production (nondecision time; nDT). Importantly, different signatures of brain activity were shown to reflect distinct aspects of this mechanistic model, and therefore, single-trial measurements of the relevant brain activity can be used to constrain these models based on the underlying neural processes[17–22].

To date, few studies have exploited such neural markers of dissociable representations associated with sensory and decision evidence to arbitrate between different accounts of how multisensory evidence influences decisions in the human brain[23]. While some studies have performed careful comparisons between diffusion models and behavioral data[24–27], they did not constrain these models against neural activity. Other studies, in contrast, tried to dissociate pre- and post-perceptual mechanisms by traditional activation mapping, but without a clear mechanistic model reflecting the decision process to support the interpretation of brain activity[28–31]. Furthermore, many studies focusing on visual judgements have considered only very simplistic stimulus features, such as contrast, salience, random-dot motion, or orientation[7,32–35], which may be encoded locally at the level of early sensory processing, and hence may not generalize to complex real-life conditions. As a result, the neural mechanisms governing the influence of information from one modality on the decision-making process of another modality remain unknown.

In this work, we employ a well-established visual object categorization task, in which early sensory evidence and post-sensory decision evidence can be properly dissociated based on electroencephalography (EEG) recordings. Specifically, using a face-vs-car categorization task, we have previously profiled two temporally distinct neural components that discriminate between the two stimulus categories: an early component, appearing ~170–200 ms poststimulus onset, and a late component, seen after 300–400 ms following the stimulus presentation[36–41]. In this previous work, we found that the late component was a better predictor of behavior than the early component, as it predicted changes in the rate of evidence accumulation in a DDM and

shifted later in time with longer deliberation times[36,42–44]. Taken together, these findings established that the early component encodes the initial sensory evidence, while the late component encodes post-sensory decision evidence.

Here, we capitalized on these distinct validated neural representations of visual information to identify the stage at which complimentary auditory information influences the encoding of decision-relevant visual evidence in a multisensory context. Based on recent results[9–11,40], we hypothesized that using AV information to discriminate complex object categories—rather than more primitive visual features—would lead primarily to enhancements in the Late, as opposed to the Early, component, consistent with a post-sensory account. Importantly, by combining single-trial modelling and EEG data, we exploited the trial-by-trial variability in the strength of the Early and Late neural components in a neurally informed DDM to derive mechanistic insights into the specific role of these representations in decision-making with AV information. In short, we demonstrate in this work that multisensory behavioral improvements in accuracy arise from enhancements in the quality of post-sensory, rather than early sensory, decision evidence, consistent with the emergence of multisensory information in higher-order brain networks.

## Results

**Behavioral performance.** We collected behavioral and EEG data from 40 participants during a speeded face-vs-car categorization task (Fig. 1). Participants were required to identify a noisy image as being either a face or a car, presented in a randomly interleaved fashion either alone (visual trials; V trials) or simultaneously with distorted speech or car sounds (audiovisual trials; AV trials) for 50 ms. The amount of visual evidence (image phase coherence) varied consistently across participants over four levels, whereas the quality of the auditory evidence (distortion level) was set at a participant-specific level throughout the task. This level was determined by calculating the amount of distortion required for correct discrimination of 68–72% of trials during an auditory-only training session on the previous day (see "Methods").

We used generalized linear mixed-effects models (GLMMs) and post hoc likelihood-ratio ($\chi^2$) model comparisons to evaluate decision accuracy and response times (RTs; using a binomial logit and a gamma model, respectively), both as a function of modality

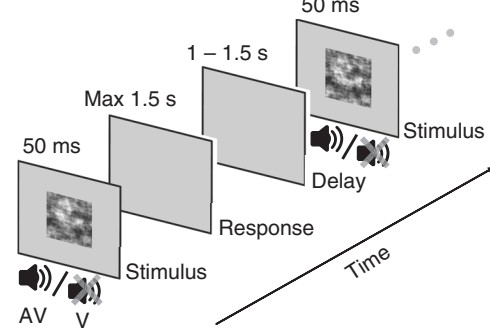

**Fig. 1 Experimental paradigm.** Schematic representation of the task design illustrating the order of presented events on the testing day. Participants had to categorize noisy representations of faces and cars. A brief stimulus, which was either an image (V) or a congruent image and sound (AV), was presented for 50 ms and followed by a delay period of up to 1500 ms during which participants were required to indicate their decision with a button press. Their response was followed by an intertrial interval (blank gray screen), jittered between 1000 and 1500 ms in duration, before the next stimulus was presented.

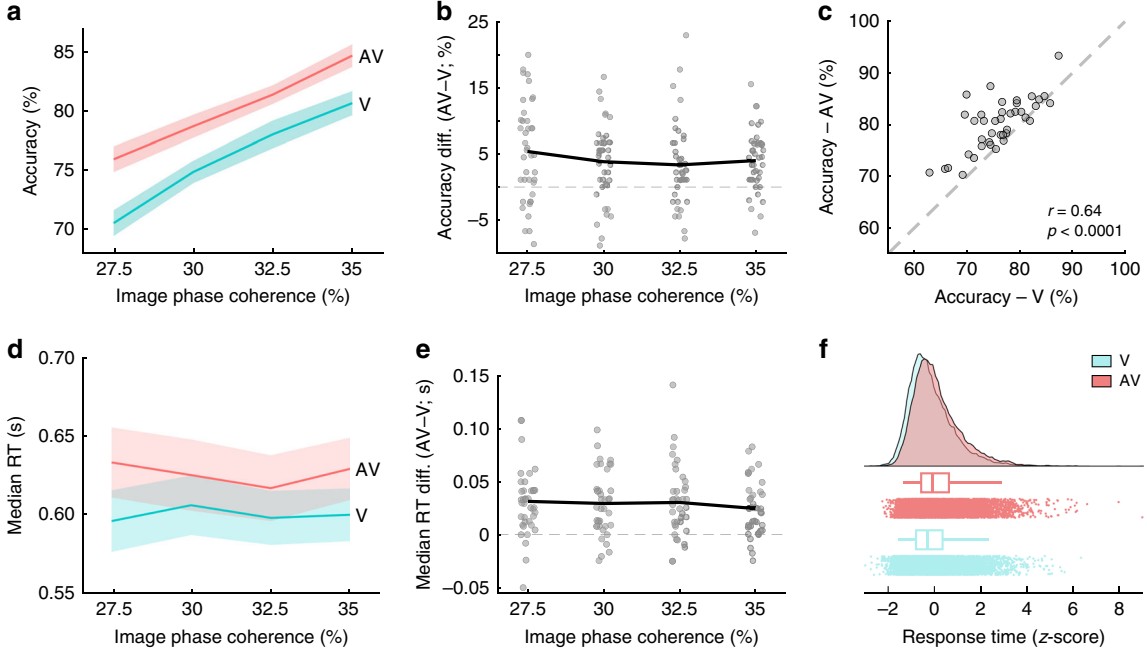

**Fig. 2 Behavioral performance. a, d** Group averages of **a** decision accuracy (mean) and **d** response time (RT; median) across the four levels of visual evidence (phase coherence) and as a function of the visual (V; turquoise) and audiovisual (AV; red) trials. Shaded error bars indicate standard errors of the mean (SEM) and median across participants ($n = 40$), respectively. **b, e** Individual participant behavioral performance changes (AV–V trials) for **b** decision accuracy and **e** RT across the four levels of visual evidence (phase coherence). Group averages were computed across $n = 40$ independent participants. Solid black lines indicate group averages. **c** Robust bend correlation between individual participant decision accuracy, computed across all four levels of visual evidence (one value per participant), during V and AV trials. Bending (i.e., down-weighting) was performed on 20% of the data points in each direction. Dashed gray line represents equal performance during V and AV trials. Indicated correlation statistics were obtained from a robust bend correlation. **f** Standardized RTs of single trials. RTs were standardized on the participant level. Dots of the raincloud plot denote single-trial RTs[108]. Source data for this figure are provided as a Source data file.

(V/AV) and the levels of visual evidence (see "Methods"). We find that participants perform more accurately during AV than during V trials ($\chi^2 = 30.02$, df = 1, $p < 0.001$; Fig. 2a–c), as well as with increases in the amount of visual evidence ($\chi^2 = 204.51$, df = 3, $p < 0.001$; Fig. 2a, b). Our data shows no significant interactions between modality and the level of visual evidence ($\chi^2 = 0.60$, df = 1, $p = 0.4376$; $\chi^2 = 0.01$, df = 1, $p = 0.9142$; $\chi^2 = 0.69$, df = 1, $p = 0.4047$, respectively; sorted by increasing coherence level), and very strong evidence for an alternative model without interactions given our data from a Bayesian mixed model analysis ($BF_{10} = 5030.05 \pm 0.62\%$; see "Methods"). RTs increase somewhat with AV evidence ($\chi^2 = 18.78$, df = 1, $p < 0.001$) and decrease with the amount of visual evidence ($\chi^2 = 48.71$, df = 3, $p = 0.0011$; Fig. 2d, e). The RT model shows no significant interactions between modality and the level of visual evidence ($\chi^2 = 1.43$, df = 1, $p = 0.2322$; $\chi^2 = 1.53$, df = 1, $p = 0.2156$; $\chi^2 = 0.004$, df = 1, $p = 0.9522$), and very strong evidence for an alternative model without interactions given our data ($BF_{10} = 1848.15 \pm 0.53\%$).

To ensure that our choice in the amount of participant-specific auditory evidence could not independently explain the overall improvements in accuracy during AV trials, we quantified the extent to which participants provided with higher levels of auditory evidence benefited more in AV trials. We find that the amount of auditory evidence explains only a minimal fraction of the variance in accuracy across participants ($R^2 = 0.01$). In addition, we find that participants who perform well in V trials also perform well in AV trials ($r_{bend}(38) = 0.64$, $p < 0.0001$), and the majority of participants (90%) show improved decision accuracy with additional auditory evidence (Fig. 2c).

The nature of our experimental design (i.e., learning to associate short sounds with specific visual categories) may have encouraged some participants to adopt a strategy in which—in some trials—they only used the complementary auditory information when the visual evidence alone did not allow them to categorize the stimulus (rather than a consistent combination of both pieces of evidence). It follows that in this subset of trials, RTs would increase leading to a bimodality in the RT distribution, which could have been concealed by group differences. To rule this out, we first standardized each participant's RTs (by z-scoring) and then tested the resulting distributions for bimodality, using a mixture of one or two exponentially modified Gaussian distributions (see "Methods"). We find that one exponential Gaussian fits our RT data best (BIC = −1064 vs −981 for V and BIC = −912 vs −688 for AV; Fig. 2f).

Taken together, these results suggest that the combined influence of audiovisual information indeed contributes to an increased likelihood of making a correct decision (overall improvement $M = 4.14\%$, standard deviation (SD) = 3.91%), but at the cost of response speed (overall slowing across visual coherence levels $M = 33.1$ ms, SD = 35.02 ms). The latter is likely due to additional time required for encoding the auditory stimulus (see "Discussion").

**Time course of the impact of sounds on visual representations.** Next, we analyzed the EEG data to identify temporally distinct components that discriminate between face and car stimulus categories. We performed this analysis separately for V and AV trials to characterize the extent to which the visual

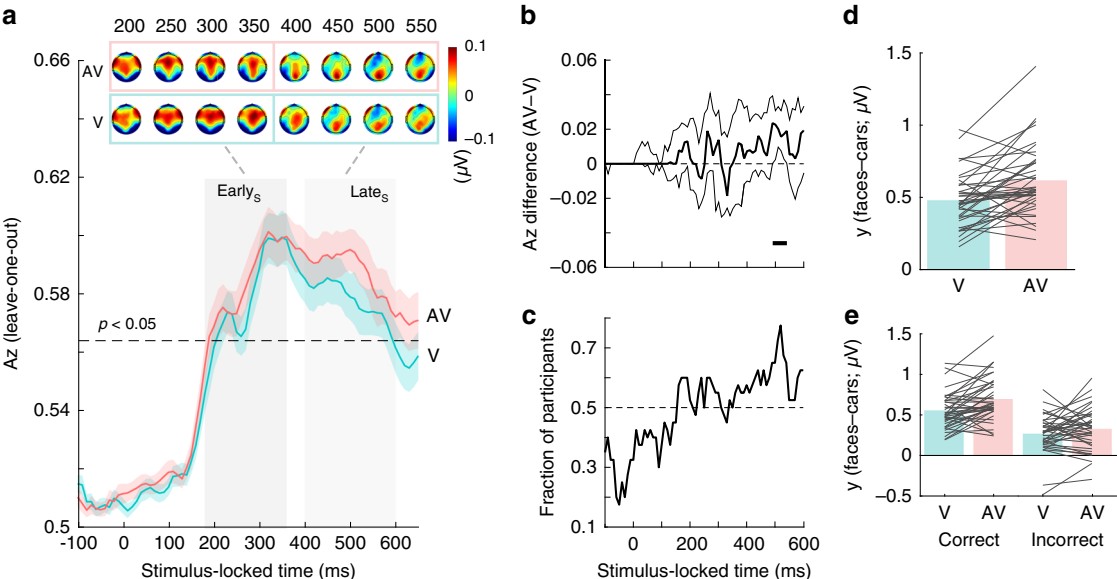

**Fig. 3 Stimulus-locked face-vs-car discrimination analysis. a** Mean discriminator performance ($A_z$) during face-vs-car discrimination of stimulus-locked EEG data after a leave-one-trial-out cross-validation procedure, as a function of the visual (V; turquoise) and audiovisual (AV; red) conditions. Dashed black line represents the group average permutation threshold at $p < 0.05$. Shaded error bars indicate bootstrapped standard errors (SEM) across participants. Shaded gray vertical bars indicate Early and Late EEG component windows determined using temporal clustering of scalp topographies. The number of clusters and their extent was obtained via a $k$-means clustering algorithm that used a Euclidean distance metric and optimized $k$ (for details, see "Methods"). These bars do not indicate statistical significance. Scalp topographies at representative time windows corresponding to the Early and Late EEG components, encoding sensory and post-sensory visual evidence, respectively. Colourbar represents normalized amplitude ($\mu V$). **b** Bootstrapped difference in discriminator performance (AV−V; thick black line) with 95% confidence intervals (2.5–97.5%; thin black lines). Horizontal thick black line above the $x$-axis in **b** illustrates significant temporal windows resulting from this one-sided permutation test (i.e., those in which the lower confidence interval is greater than zero ($p < 0.025$) with an added data-driven minimum requirement of three contiguous windows to correct for multiple comparisons; see "Methods"). **c** Fraction of participants showing discriminator performance ($A_z$) in the same direction as the group-level mean. **d** Late EEG component amplitudes reflecting the relative separation across all face and car trials ($y_{faces}-y_{cars}$) at the point of maximum $A_z$ separation between V and AV trials (see **a, b**). **e** Late EEG component amplitudes ($y$) as in **d** separated by decision accuracy. Reproducibility was ensured using an out of sample leave-one-trial-out cross-validation procedure per participant, thereby each participant becomes its own replication unit (for details, see "Methods"). Source data for this figure are provided as a Source data file.

representations encoded in these components were affected by the additional auditory evidence. Specifically, for each participant separately, we performed a single-trial multivariate discriminant analysis[45,46] to estimate linear spatial weights (i.e., spatial filters) that maximally discriminated face-vs-car trials within short pre-defined temporal windows, locked either to the onset of the stimulus or the response (see "Methods").

Applying the resulting spatial filters to single-trial data produces a measure of the discriminating component amplitudes (henceforth $y$), which can be used as an index of the quality of the visual evidence in each trial[36,42,47,48]. In other words, more extreme amplitudes, positive or negative, indicate more face or car evidence respectively, while values closer to zero indicate less evidence. To quantify the discriminator's performance over time and identify the relevant components, we used the area under a receiver operating characteristic (ROC) curve (henceforth $A_z$ value) with a leave-one-trial-out cross-validation approach to control for overfitting.

The discriminator's performance as a function of stimulus-locked time reveals a broad window over which face-vs-car decoding was statistically reliable for both V and AV conditions (i.e., 180–600 ms poststimulus; Fig. 3a). To identify the number of relevant components in this range, we applied temporal clustering on the resulting scalp topographies as in previous work (see "Methods")[41]. This procedure reveals the presence of two temporally distinct scalp representations with a transition point at 380 ms poststimulus for both the V and AV conditions. These

spatial representations are consistent with our previously reported Early and Late components, with centrofrontal and bilateral occipitotemporal activations for the Early and a prominent centroparietal activation cluster for the Late component (Fig. 3a, top)[36–38,42,43].

We then extracted participant-specific component latencies—for each condition separately—by identifying the time points leading to peak $A_z$ performance within each of the two windows identified by the clustering procedure. We allowed a 40 ms gap centered on the transition point to avoid potential multiplexing effects (i.e., we considered stimulus-locked windows 180–360 ms and 400–600 ms, for the Early and Late components, respectively). The mean peak times for the Early component for the V and AV conditions are 293 ms ($SD_V = 53.84$ ms, $SD_{AV} = 57.52$ ms). The mean peak times for the Late component are 500.25 ms ($SD_V = 40.92$ ms) and 508.25 ms ($SD_{AV} = 40.76$ ms) for the V and AV conditions, respectively. Our data shows no statistically significant latency differences across V and AV conditions (Early: two-sided paired $t$ test, $t(39) = 0.00$, $p = 1$; Late: two-sided paired $t$ test, $t(39) = -1.09$, $p = 0.281$).

Moreover, the seemingly separate peaks in the discriminator performance (Fig. 3a) within the earlier temporal window (180–360 ms poststimulus) are likely due to interindividual differences in the onset of the Early component (i.e., differences in early sensory encoding). We tested this formally by demonstrating that the distributions of the Early component peak latencies are best approximated by a mixture of two—rather

than one—Gaussians (BIC = 433 vs 438 for V and BIC = 440 vs 444 for AV) with means ~240 and ~330 ms, respectively, which coincide with the two peaks in the aforementioned window.

Our main goal in this work is to determine when the decision-relevant category information is enhanced in the AV condition. We, therefore, sought to identify temporal windows during which the discriminator performance differs systematically between V and AV trials, and test the extent to which they overlap with the Early and/or Late components. Specifically, we used a temporal cluster-based permutation analysis, whereby for each temporal sample we created a bootstrap distribution of group-level $A_z$ difference scores (AV−V) and compared the bootstrapped median difference score against the lower bound of the estimated confidence interval of the distribution (supporting a significance level of $p < 0.025$)[49,50]. To form contiguous temporal clusters and avoid transient effects due to false positives, we required a data-driven minimum temporal cluster size of at least three significant samples (see "Methods").

This analysis shows only a single temporal cluster overlapping with the Late component (490–540 ms) over which the discriminator performance for AV trials is significantly improved compared to V trials (Fig. 3a, b). During this time, up to 78% of participants exhibited increases in the discriminator's performance for AV trials, compared to only 60% of participants during the Early component (Fig. 3c). These findings indicate that the addition of auditory information in our task enhances primarily the quality of visual evidence (as reflected in our discriminator component amplitudes $y$) during post-sensory decision-related processing (Fig. 3d). This enhancement of the quality of decision evidence during AV trials is comparable across both correct and incorrect trials—with the quality of evidence being overall higher during correct compared to incorrect trials (Fig. 3e; $BF_{10} = 10.27 \pm 0.9\%$). Further, our data provide evidence against an interaction between AV benefit and decision accuracy ($BF_{10} = 2.06 \pm 1.3\%$).

To rule out that these post-sensory enhancements are not driven by the speed (and hence the efficiency) with which participants encode the early sensory evidence, we performed two complementary analyses. Specifically, we correlated our Early component peak times with (1) the differential Late component amplitude effects (AV−V) and (2) the peak times of the Late component. In both analyses, we observe that the latency of the Early component has no significant leverage on the neural correlates of the Late component ($r_{bend} = -0.28$, $p = 0.071$ and $r_{bend} = 0.22$, $p = 0.1752$, respectively).

In previous work, we showed that the Late component activity starts out as being stimulus-locked, but persists and becomes more robust near the response[36,41,42], consistent with the notion that decision evidence reverberates and accumulates continuously until one commits to a choice. We therefore repeated the single-trial multivariate discrimination analysis on response-locked data. Importantly, this analysis also helps to rule out potential motor confounds associated with differences in RTs across V and AV trials by abolishing potential temporal lags near the time of the response.

As with the stimulus-locked analysis, we compared the face-vs-car discriminator performance between V and AV trials. This analysis reveals a temporal cluster leading up to the eventual choice (−110 to −60 ms pre-response) during which discriminator performance is significantly enhanced for AV compared to V trials (Fig. 4a, b), with consistent effects (>70%) appearing across participants (Fig. 4c). Inspection of the resulting scalp maps during this period indicates that the spatial topographies, featuring a prominent centroparietal cluster, are consistent with the Late component seen in the stimulus-locked analysis ($r_{bend}(38) = 0.88$ for V and 0.86 for AV; compare scalp topographies for $Late_S$ and $Late_R$ in Figs. 3a and 4a), in line

with previous work[19,41,44]. These findings further highlight that it is primarily late, decision-related visual evidence that is being amplified during audiovisual object categorization (Fig. 4d). Similar to the stimulus-locked Late component, we find very strong evidence that this amplification in AV trials arises independently of the accuracy of the decision, while overall neural evidence is higher for correct trials (Fig. 4e; $BF_{10} = 506.3 \pm 0.73\%$). There is no interaction between this amplification of neural evidence and the accuracy of a decision (Fig. 4e; $BF_{10} = 3.05 \pm 0.86\%$).

**Neurally informed modelling explains multisensory effects.** Having characterized whether the added influence of auditory information enhances early sensory or late post-sensory visual representations, we then asked whether the identified single-trial neural responses are directly linked to improvements in behavior between V and AV trials. To this end, we employed a neurally informed variant of the traditional hierarchical drift diffusion model (HDDM; see "Methods"), a well-known psychological model for characterizing rapid decision-making[14,51,52] to offer a mechanistic account of how the human brain translates the relevant evidence into a decision. In doing so, we directly constrained the model based on additional neural evidence, hence closing this persistent gap in the literature[26,28,30].

In brief, the traditional HDDM decomposes task performance (i.e., choice and RT) into internal components of processing representing the rate of evidence integration (drift rate, $\delta$), the amount of evidence required to make a choice (decision boundary separation, $\alpha$), the duration of other processes, such as stimulus encoding and response production (nDT), and a potential bias or prior information favouring one of the two choices (starting point, $\beta$). Ultimately, by comparing the obtained values of all these HDDM parameters across the V and AV trials, we could associate any behavioral differences resulting from the addition of auditory information (improved performance and longer RTs as in Fig. 2) to the constituent internal processes reflected by each of these parameters.

Importantly, we deployed a neurally informed HDDM (nHDDM), whereby we incorporated single-trial EEG component amplitudes ($y$-values) into the parameter estimation (Fig. 5a). Specifically, we extracted single-trial discriminator amplitudes from participant-specific temporal windows (i.e., peak $A_z$ difference across AV−V) corresponding to both the Early and the Late stimulus-locked EEG components (see "Methods"). Since these values represent the amount of face or car evidence available for the decision (i.e., indexing the quality of the visual evidence as we demonstrated in previous work[38,42,43]), we used them to construct regressors for the drift rate parameter in the model ($\gamma_{Early}$, $\gamma_{Late}$), based on the idea that evidence accumulation is faster when the neural evidence for one of the choices is higher. We therefore estimated these regression coefficients ($\gamma_{Early}$, $\gamma_{Late}$) to directly assess the relationship between trial-to-trial variations in EEG component amplitudes and drift rate.

We further hypothesized that the reliability of sensory information (as reflected by the visual coherence levels) would affect the rate of information integration. Thus, as per common practice[14,53], we modeled a linear relationship between drift rate and coherence levels. To investigate whether this relationship is modulated by the Early and/or Late EEG component amplitudes, we tested three models where coherence scaled (a) $y_{Early}$, (b) $y_{Late}$, or (c) both components. We find the best fit for the model where coherence scales $y_{Late}$ (deviance information criterions (DICs) = 767, 517, and 661, respectively), indicating that the modulation of the Late component with the reliability of available evidence predicts the rate of evidence accumulation. In other words, the

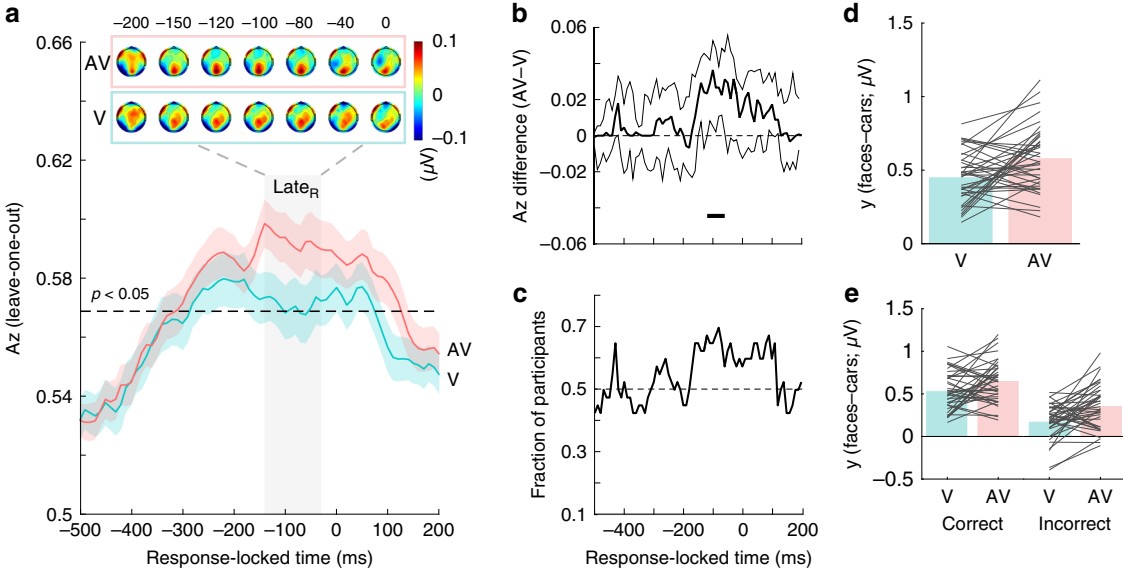

**Fig. 4 Response-locked face-vs-car discrimination analysis. a** Mean discriminator performance ($A_z$) during face-vs-car discrimination of response-locked EEG data after a leave-one-trial-out cross-validation procedure, as a function of the visual (V; turquoise) and audiovisual (AV; red) conditions. Dashed black line represents the group average permutation threshold at $p < 0.05$. Shaded error bars indicate bootstrapped standard errors (SEM) across participants. Shaded gray vertical bar indicates Late component window used in the selection of component amplitudes ($y$) shown in **d**. This window resulted from a temporal cluster-based bootstrap analysis performed on median $A_z$ difference scores (for details, see description of **b** and "Methods" section). Scalp topographies at representative time windows corresponding to the Late EEG component (indicated by dashed lines) encoding persistent post-sensory visual evidence up until the eventual commitment to choose. Colourbar represents normalized amplitude ($\mu V$). **b** Bootstrapped difference in discriminator performance (AV−V; thick black line) with 95% confidence intervals (2.5–97.5%; thin black lines). Gray shaded area in **a** and horizontal thick black line above the x-axis in **b** illustrate significant temporal windows resulting from this one-sided permutation test (i.e., those in which the lower confidence interval is greater than zero ($p < 0.025$) with an added data-driven minimum requirement of three contiguous windows to correct for multiple comparisons; see "Methods"). This procedure corrects for multiple comparisons. **c** Fraction of participants showing discriminator performance ($A_z$) in the same direction as the group-level mean. **d** Late EEG component amplitudes reflecting the relative separation across face and car trials ($y_{faces}$−$y_{cars}$) at the point of maximum $A_z$ separation between V and AV trials (see **a**, **b**). **e** Late EEG component amplitudes ($y$) as in **d** separated by decision accuracy. Reproducibility was ensured using an out of sample leave-one-trial-out cross-validation procedure per participant, whereby each participant becomes its own replication unit (for details, see "Methods"). Source data for this figure are provided as a Source data file.

best fitting model suggests that the effect of task difficulty on behavioral performance is captured by post-sensory mechanisms. Critically, this result dissociates the roles of the Early and Late EEG components in the decision-making process and is consistent with the role of the Late component in indexing the quality of the evidence entering the decision process (as has been shown in past work[38,42,43]), which, unlike early sensory encoding, is more closely associated with the accuracy of perceptual choices.

When applying this model to the behavioral data, we obtain a good fit, accounting for most of the variance in the choice and RT data (average $R^2 = 0.94$; Fig. 5b). Consistent with the functional role of the Early and Late EEG components in conveying sensory and post-sensory evidence, respectively, the within-participant single-trial discriminator amplitudes of both components are predictive of drift rate in both sensory conditions (Fig. 5c, d; $\gamma_{Early}$ and $\gamma_{Late}$ significantly larger than zero for both V and AV, $t(39)$ = 17.67, $t(39)$ = 15.55 for $\gamma_{Early}(V)$, $\gamma_{Early}(AV)$, respectively, and $t(39)$ = 11.92, $t(39)$ = 16.02 for $\gamma_{Late}(V)$, $\gamma_{Late}(AV)$, respectively, all $p$ values < 0.001; all two-sided paired $t$ tests). Our results also show that the drift rates of correct trials are on average higher than those of incorrect trials ($\delta_{correct} = 0.27 \pm 0.09$, $\delta_{incorrect} = 0.14 \pm 0.11$ for face choices and $\delta_{correct} = -0.65 \pm 0.11$, $\delta_{incorrect} = -0.34 \pm 0.10$ for car choices—with the convention of positive signs for faces and negative signs for cars). This finding provides strong support that accuracy effects are effectively directly captured by our nHDDM.

Crucially, the contribution of the Late but not the Early component (i.e., $\gamma_{Late}$, but not $\gamma_{Early}$) is significantly higher in AV compared to V trials (Fig. 5d; two-sided paired $t$ tests: $t(39) = -0.6891$, $p = 0.4984$ for $\gamma_{Early}$, $t(39) = -2.66$, $p = 0.011$ for $\gamma_{Late}$). This is consistent with the increased discrimination power of the Late component in AV trials, and suggests that this component underpins the behavioral facilitation of evidence accumulation via post-sensory amplification of the available decision evidence (via the added auditory information) entering the decision process.

We subsequently investigated the effect of the additional auditory information on the three other parameters of the nHDDM. Our data shows no reliable difference in starting point and boundary separation between the two sensory conditions ($\beta_V = 0.5475 \pm 0.005$, $\beta_{AV} = 0.5495 \pm 0.005$; two-sided paired $t$ test: $t(39) = -0.4964$, $p = 0.6224$; Fig. 5e and $\alpha_V = 1.13 \pm 0.03$, $\alpha_{AV} = 0.12 \pm 0.03$; two-sided paired $t$ test: $t(39) = 0.8191$, $p = 0.4177$; Fig. 5f), and significantly longer nDTs during AV trials ($370 \pm 9$ ms for V vs $408 \pm 10$ ms for AV; two-sided paired $t$ test: $t(39) = 2.81$, $p = 0.0063$; Fig. 5g). The latter result is likely related to longer stimulus encoding in AV trials, which may result from the extra time required to process the auditory stimulus (see "Discussion"). Notably, the average difference in RTs (33 ms) is comparable with the average nondecision difference between the two conditions (38 ms), which provides further evidence for the early sensory origins of the longer RTs in AV trials.

**Neurally informed modelling of choice biases.** Next, we explored additional analyses that had no direct impact on the multisensory effects reported above, but nonetheless captured relevant idiosyncratic strategies in choice behavior. Specifically,

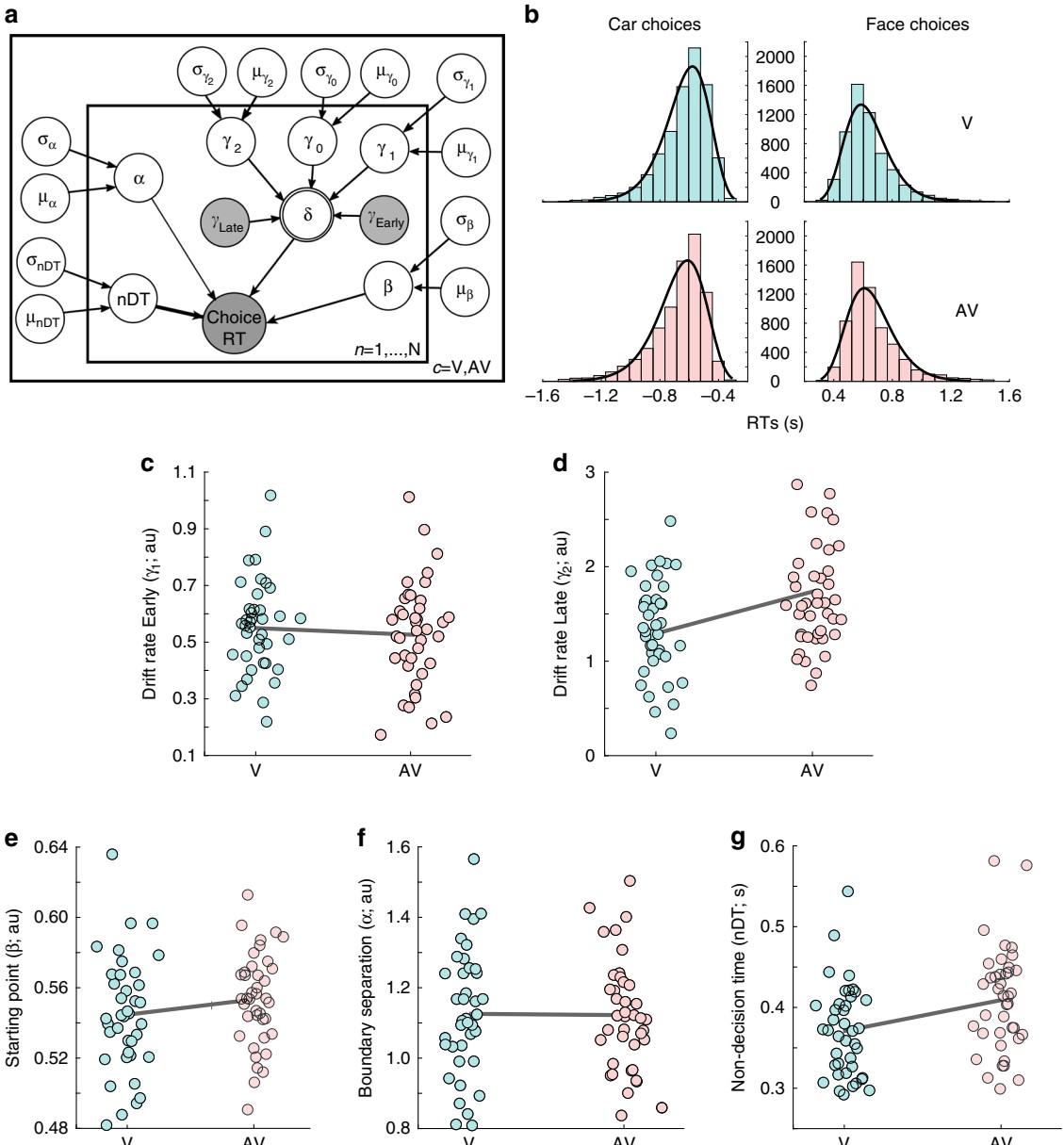

**Fig. 5 Neurally informed cognitive modelling. a** Graphical representation showing hierarchical estimation of nHDDM parameters. Round nodes represent continuous random variables and double-bordered nodes represent deterministic variables, defined in terms of other variables. Shaded nodes represent recorded or computed signals, i.e., single-trial behavioral data (choice, RT) and EEG component amplitudes ($y$'s). Parameters are modeled as random variables with inferred means $\mu$ and variances $\sigma^2$. Plates denote that multiple random variables share the same parents and children. The outer plate is over sensory conditions (V, AV) and the inner plate is over participants ($n$). **b** Histogram and nHDDM model fits for RT distributions of car (left) and face (right) choices in the visual (V; top—in turquoise) and audiovisual (AV; bottom—in red) conditions. **c, d** Regression coefficients ($\gamma$) of the **c** Early and **d** Late EEG component amplitudes ($y$'s) in V (turquoise) and AV (red) conditions, as predictors of the drift rate ($\delta$) of the nHDDM shown in **a**. Coefficients derived from nHDDM, including $n = 40$ independent participants and 28,540 trials. **e** Starting point values ($\beta$) estimated by the nHDDM for V (turquoise) and AV (red) conditions. **f** Boundary separation values ($\alpha$) estimated by the nHDDM for V (turquoise) and AV (red) conditions. **g** Nondecision times (nDT) estimated by the nHDDM for V (turquoise) and AV (red) conditions. Dots indicate single-participant values and gray lines connect the population means in **c**–**g**. Source data for this figure are provided as a Source data file.

we observe a starting point bias closer to face choices (expressed as a proportion of the boundary separation, two-sided paired $t$ tests: $t(39) = 9.06$ for V and $t(39) = 10.63$ for AV, both $p$ values < 0.001) and a higher drift rate for car choices, in both the V and AV conditions ($\delta_{car-V} = -0.92 \pm 0.14$, $\delta_{face-V} = 0.15 \pm 0.13$, $\delta_{car-AV} = -0.96 \pm 0.14$, $\delta_{face-AV} = 0.07 \pm 0.12$—positive (negative) signs indicate face (car) choices). To understand these results, we examined potential differences in the behavioral results between face and car choices. In particular, participants choose cars more

often than faces (60 ± 8% of the V trials and 52 ± 8% of the AV trials were car choices), and are more accurate (84% vs 74% on average) in trials with a face choice. This accuracy effect is comparable across V and AV trials (two-sided paired $t$ test: $t(39) = -0.98$, $p = 0.3353$; 83.2 ± 1.34% for V and 86.59 ± 1.14% for AV face choices, 72.84 ± 0.98% for V and 76.9 ± 0.91% for AV car choices).

These results suggest that the bias in the starting point is likely driving the accuracy difference between face and car choices (i.e.,

more (fewer) errors when the starting point is further from (closer to) the correct boundary), while the higher number of car responses could be explained by a higher drift rate during car choices. The latter is also consistent with the RT distributions for face and car choices (Fig. 5b), where car RTs appear to have slightly longer tails but modes similar to face RTs. Importantly, these two parameter differences are consistent across the two conditions (V and AV), and thus they have no impact on the behavioral multisensory effects and their underlying neural mechanisms.

Finally, given the timing of the Early component, we also considered whether it relates to the duration of sensory processing mechanisms captured in the nHDDM by nDTs. Thus, we also tested a model with $y_{Early}$ as a regressor for nDT (rather than drift rate). However, this model demonstrates no relationship between $y_{Early}$ and nDT (regression coefficients are not significantly different from 0 for both V and AV). Moreover, this model yields a poorer fit of the data compared to the model of choice (DIC = 1277 vs 517 for the chosen nHDDM). This finding is also consistent with the notion that increases in the nDTs observed in AV trials are likely driven by increases in the early encoding time of the added auditory information.

**Neurally informed model outperforms behaviorally constrained model.** Given that most previous studies in multisensory decision-making have fit DDMs only to behavioral data, it is worth asking whether the inclusion of EEG-derived regressors actually improves model performance and/or shapes the conclusion derived from the model. We formally compared the neurally inspired HDDM to a standard HDDM without neurally informed constraints. The traditional model yields a poorer trade-off between goodness-of-fit and complexity, as assessed by the DIC for model selection[54], compared to its neurally informed counterpart ($DIC_{HDDM} = 758$ vs $DIC_{nHDDM} = 517$). In addition, the conclusions that would have been derived from such a poorer model contradict those reported above. For example, the conventional HDDM yields larger boundary separations for AV trials (two-sided paired $t$ test: $t(39) = -3.52$, $p = 0.0011$), the nDTs estimated by this model are ~100–120 ms longer for both sensory conditions compared to the nHDDM ($490 \pm 10$ ms for V and $509 \pm 11$ ms for AV), and the difference in average nDTs across conditions (19 ms) does not track the mean RT difference as closely as the nDTs estimated by the nHDDM. Hence, this poorer performing model constrained only on the behavioral data could lead to the misleading conclusion that the auditory information also affects the response caution (or individual speed-accuracy trade-off strategies via boundary adjustments). This supports the importance of constraining behavioral models with neural data and suggests that integrating neural information in these models can potentially enable a more accurate characterization of the behavioral effects, as well as a mechanistic interpretation of their neural correlates.

## Discussion
In this work, we used multivariate single-trial EEG analysis and behavioral modelling to investigate the enhancement of visual perceptual decisions by complementary auditory information. We showed that significant improvements in behavioral performance in AV trials were accompanied primarily by enhancements in a late EEG component indexing decision-related processes[36–38,42,43]. In contrast, an earlier EEG component reflecting sensory (visual) evidence remained unaffected by the addition of auditory evidence. Using neurally informed cognitive modelling, we showed that these multisensory behavioral and

neural benefits could be explained primarily by improvements in the rate of evidence accumulation in the decision process itself.

The processing of multisensory information requires the coordination of multiple mechanisms serving bottom-up, top-down, and predictive coding processing[3]. The neural implementation of these mechanisms involves a distributed neural network, including primary sensory, parietal, and frontal brain areas that interact with each other to form and shape multisensory perception[10,11].

In this study, we were particularly interested in when multisensory information is combined to improve perceptual judgements. In the field of multisensory decision-making, there are two prominent theories that emphasize either the role of early or late integration of multisensory information, respectively[2]. The early integration hypothesis[55–57] posits that sensory evidence is combined at the stage of early sensory encoding. This hypothesis is supported by evidence for direct pathways between early visual and auditory regions, or cross-modal influences on neural responses in early visual cortices[55,58–62] and studies demonstrating benefits for the perception of simplistic visual stimuli, such as contrast[7,63], motion direction[5,32], and simple shape discrimination[29] from acoustic information. However, the use of such simple stimuli may have specifically engaged only early sensory regions, hence providing a biased interpretation that does not generalize to more complex objects.

In contrast, the late integration hypothesis, postulates that evidence from each sensory modality is instead processed separately during early sensory encoding, and is combined into a single source of evidence downstream, during the process of decision formation itself[2]. Support for this hypothesis comes from both animal and human experiments demonstrating that multisensory information is accumulated right up to the point of a decision, while processing of unisensory information occurs prior to the formation of a multisensory decision[8,64]. Similarly, recent neuroimaging work has provided new insights that flexible behavior can be accounted for by causal inference models[65], with multisensory representations converging on higher-level parietal and prefrontal regions (e.g., inferior parietal sulcus, superior frontal gyrus) previously linked to the process of evidence accumulation[9–11,66,67].

Our findings appear to be at odds with the early integration hypothesis, since we found no evidence that the addition of auditory information had any impact on the encoding of early visual evidence, which remained comparable between V and AV trials. Instead, we offered support for post-sensory enhancements of decision evidence with the addition of auditory information that is most consistent with the late integration hypothesis. Importantly, these later visual representations are likely to reside in higher-order visual areas involved in object recognition and categorization (e.g., lateral occipital cortex), as we have shown previously[38], consistent with the emergence of multisensory evidence only after early sensory encoding[9]. Specifically, the timing of these representations (starting after early sensory encoding and lasting until the commitment to choice) suggests that they unfold concurrently with the decision and provide the input to the process of evidence accumulation in prefrontal and parietal cortex[66–70].

A potential confounding factor for the late multisensory effects observed in our data could be differences in attention between V and AV conditions. If such unspecific effects were indeed at play, they would have likely impacted both early and late processing stages in a similar manner. Moreover, recent work suggests that the influences of multisensory information and attention operate independently across cortical columns[71], and that attentional resources are largely shared across sensory modalities[72]. Hence, arguing against a competition between sensory modalities for

attentional resources. Experimentally, we fully randomized all trials to ensure participants were equally likely to encounter (and expect) V or AV stimuli during each trial, thereby minimizing differences in attention between sensory conditions.

Another potential point of departure from previous studies is that we observed increased RTs during multisensory trials. This finding likely stems from the auditory information being context-dependent and complementary to the visual information, rather than redundant as in previous work[73]. In other words, the sounds in our task are treated as supplementary evidence, instead of simply providing confirmation of the visual evidence, that require the deployment of additional processing resources, consistent with the observed increases in nDTs for multisensory trials in our modelling results.

An alternative interpretation of these increases in nDT during multisensory trials provoked by a lack of latency differences in our Early (sensory) component across V and AV trials could be differences in motor preparation. This interpretation is highly unlikely because participants indicated their decision, using the same motor effectors and button press in both V and AV trials. Furthermore, our hypothesized increases in sensory encoding time due to the additional processing of auditory evidence during AV trials would not have been reflected in the latency of our Early component, which remained unaffected by the presence of the additional auditory evidence.

Correspondingly, reaction time differences could arise due to the particular choice of sensory modalities and/or interindividual choice strategies employed by participants. For example, a recent study[26] using time-varying multisensory information (visual and vestibular) reported faster, but slightly less accurate choices for multisensory compared to unisensory decisions. The authors modeled these results with a variant of the DDM model that incorporates the effects of time-varying information and sensory cue reliability and reported consistent drift rate improvements in the multisensory condition across participants. In other words, despite differences in the behavioral outcomes, their findings are in line with the increase in drift rate in AV trials we observed in the present study; that is, both studies suggest that multisensory information leads to faster accumulation of sensory evidence.

Crucially in this work, we were able to characterize the neural underpinnings of the behavioral benefits obtained from the addition of auditory information. This contribution was made possible by the joint cognitive modelling of behavioral and neural data that linked the neural correlates of sensory and decision evidence with the internal processes involved in decision-making. Our neurally informed DDM indicated that the improvement in behavioral performance is derived mainly from enhanced post-sensory representations that modulate the rate of evidence accumulation. This result ran contrary to the behavioral-only version of a standard DDM, which attributed the longer RTs in AV trials to additional changes (increases) in the decision boundary and to a lesser extent to early encoding of the auditory stimulus.

We suggest that the reason for this discrepancy is a less accurate account of the trial-by-trial variability in the decision dynamics (also indicated by the poorer fit of the single-trial data) than its neurally informed counterpart. In other words, the inclusion of the two well-characterized EEG components provided a more accurate account of the contributions of early sensory and post-sensory decision evidence to the formation of decision dynamics. Thus, this approach enabled the disambiguation of the internal processing stages that yielded such a behavioral benefit. Additional support for this claim is provided by the fact that the behavioral model yielded longer stimulus encoding times, whose difference across conditions did not track the difference in measured RTs equally well.

Our findings suggest that constraining models of perceptual decision-making with neural data can provide key mechanistic insights, which may remain unobserved using behavioral modelling alone. This argument is in line with recent research, suggesting that the high complexity of decision-making models may yield neurally incompatible outcomes[74–76]. However, when informed by neural measurements, these models cannot only yield more reliable parameter estimates, but also shed light on the neural mechanisms underpinning behavioral effects[43,51,77–81].

It is worth noting that several previous studies have used DDMs to study multisensory decision-making. Some of these considered models in which the combination of multisensory information was explicitly hardwired, for example, to converge during sensory accumulation[25,82,83]. By doing so, these models can describe certain aspects of human behavior, but they cannot evaluate competing hypotheses about the locus of convergence. Other multisensory studies have combined behavioral modelling using DDMs and EEG, but did not use the neural data to constrain the behavioral model. Using such an approach, we have previously argued that the encoding of visual random-dot motion in early sensory regions is affected by acoustic motion[5], speaking in favor of a sensory-level integration effect. However, this sensory-level effect was not validated using an EEG-inspired DDM model, as performed here.

One explanation for these diverging findings is that the use of simpler stimuli, such as random-dot motion, may have biased the earlier study to a sensory-level effect, whereas multisensory information about more complex objects is instead combined at a post-sensory stage. This interpretation is supported by neuroimaging studies that have reported audiovisual interactions for complex stimuli mostly at longer poststimulus latencies or in high-level brain regions[84–87].

Another potentially important difference that might explain these divergent findings is the particular construction of the multisensory context across tasks. Many audiovisual integration studies use tasks in which there is a direct mapping between the source of the evidence across the two modalities, for instance, seeing a person's mouth while producing speech (i.e., lip reading) to compensate for noisy acoustic information in a bar. In the present task, as in many real-world scenarios, however, this direct audiovisual mapping is not immediately available. In our earlier example, the decision to cross the street on a foggy morning will be based on hazy objects in your visual field together with street sounds that cannot immediately be matched to individual objects. In other words, the decision to step off the curb will be based on a broader audiovisual context and a higher-level conceptualization of the evidence, such as the presence of car-like objects and sounds signaling a busy street. This is a subtle but critical distinction in deciphering the mechanisms underlying audiovisual integration and reconciling discrepancies across different experimental designs.

## Methods

**Participants**. We estimated a minimum sample size of 35 participants by an a priori power analysis for a fixed linear multiple regression model with two predictors, a medium effect size of 0.5, an alpha of 0.05, and a power of 0.95. We therefore tested 40 participants (male = 18, female = 22; mean age = 23.85, SD = 5.47) on a speeded face-vs-car categorization task. All participants were right-handed with normal or corrected-to-normal vision and no self-reported history of neurological disorders. This study was approved by the ethics committee of the College of Science and Engineering at the University of Glasgow (CSE 300150102). All participants provided written informed consent prior to participation.

**Stimuli**. We used a set of 30 grayscale images—15 of faces and 15 of cars (image size 670 × 670 pixels, 8-bits per pixel)—adapted from our previous experiments[36–38,42]. The original face images were selected from the face database of the Max Planck Institute of Biological Cybernetics[88] and car images were sourced from the Internet.

Upon retrieval of the images, the background was removed and the image placed on uniform gray background.

All images were equated for spatial frequency, contrast, and luminance, and had identical magnitude spectra (average magnitude spectrum of all images in the database). We manipulated the phase spectra of the images using the weighted mean phase technique[89], whereby we changed the amount of visual evidence in the stimuli as characterized by their percentage phase coherence. To manipulate task difficulty, we used four levels of sensory visual evidence (27.5, 30, 32.5, and 35% phase coherence). These levels were based on our previous studies[36–38,42], as they are known to result in performance spanning psychophysical threshold. Both image categories (i.e., faces and cars) contained an equal number of frontal and side views (up to ±45 degrees). We displayed all pictures on light gray background (RGB [128, 128, 128]), using the PsychoPy software[90] (version 1.83.04) for a duration of 50 ms.

Auditory sounds (15 car- and 15 face-related) were used in addition to the visually presented images in a random half of trials. Sounds were either human speech or car/street-related sounds obtained from online sources. No copyright restrictions were in place and modifications of the sound files were allowed. These were sampled at a rate of 22.05 kHz and stored as .wav files. In MATLAB (version 2015b, The MathWorks, 2015, Natick, Massachusetts), we added a 10 ms cosine on/off ramp to reduce the effects of sudden sound onsets and normalized all sounds by their SD. Subsequently, we reduced the intensity of these normalized sounds by lowering their amplitude by 80%. Sounds were embedded in Gaussian white noise, and the relative amplitude of the sounds and noise was manipulated to create 17 different levels of relative noise-to-signal ratios (ranging from 12.5 to 200% of noise relative to the lowered amplitude signal in increments of 12.5%). The resulting noisy speech- and car-related sounds were presented binaurally for 50 ms through Sennheiser stereo headphones HD 215.

The stimulus display was controlled by a Dell 64 bit-based machine (16 GB RAM) with an NVIDIA Quadro K620 (Santa Clara, CA) graphics card running Windows Professional 7 or Linux-x86_64 and PsychoPy presentation software[90]. All images were presented on an Asus ROG Swift PG278Q monitor (resolution, 2560 × 1440 pixels; refresh rate set to 120 Hz). Participants were seated 75 cm from the stimulus display, and each image subtended ~11 × 11 degrees of visual angle.

**Behavioral task**. We employed an adapted audiovisual version of the widely used visual face-vs-car image categorization task[36–38,42]. This task required participants to decide whether they saw a face or a car embedded in the stimulus. Participants were asked to indicate their decision via button press on a standard keyboard as soon as they had formed a decision. The response deadline was set at 1.5 seconds. During half of the trials, participants were also given an additional auditory cue in the form of a brief noisy sound that was congruent with the picture's content. Audiovisual face trials were accompanied by a human speech sound, whereas the audiovisual car trials were accompanied by a car-related sound, such as squeaking tires or a slammed door. All stimuli were presented for 50 ms in the center of the screen and on AV trials to both ears. Participants were explicitly instructed to pay equal attention to and base their decision on information presented in both modalities in all trials. During AV trials, pictures and sounds were presented simultaneously. More specifically, we used four levels of visual noise, but only one participant-specific auditory difficulty level, obtained at perithreshold performance during an initial auditory training task (see below). Thereby, we accounted for interindividual differences in auditory perception, independently of visual image difficulty.

This experimental paradigm required participants to attend a training and a testing session on two consecutive days at the same time of the day. On the first day (i.e., the training day), participants were asked to perform three separate simple categorization tasks to familiarize themselves with the task: (1) a visual image discrimination task (face-vs-car), (2) an auditory sound discrimination task (face/speech vs car/street sounds) and (3) an audiovisual discrimination task (face-vs-car). During the training session, participants also received visual feedback following each response (on all three tasks). Feedback was presented centrally for each of the possible three outcomes: 'Correct' written in green, 'Incorrect' written in red, and 'Too slow' written in blue (when participants exceeded the response deadline). Stimuli presentation duration for all stimuli and tasks was set to 50 ms for comparability between the training and testing days.

During the visual training task, we used the same images and all four levels of visual evidence as on the second day (i.e., the testing day). During the auditory training task, we presented sounds to participants using eight different levels of relative noise-to-signal ratios (12.5%, 37.5%, 62.5%, 93.75%, 125%, 150%, 175%, and 200% of added noise). We estimated participant-specific noise levels supporting individual perithreshold performance (i.e., ~70% decision accuracy), including levels that might have fallen in between the eight noise-to-signal ratios used in this training task (from the larger set of 17; $M = 140\%$, $SD = 45\%$). We used these individual levels for the audiovisual training task and the main experiment. During the audiovisual training task, we used all images at the four levels of visual evidence together with the participant-specific perithreshold noise level determined above. This audiovisual training task mimicked the main task presented on the second (testing) day, with the addition that participants received feedback on their choices.

Overall, on the training day, we presented 480 trials for each of the visual and auditory discrimination training tasks split into four blocks of 120 trials with a 60-second rest period between blocks. We presented 240 trials split into two blocks during the audiovisual training task. Taken together, all three training tasks lasted approximately 55 minutes on the first (training) day.

On the second day, we collected behavioral and EEG data using randomly interleaved visual (unisensory) and audiovisual (multisensory) trials in a combined task (Fig. 1). Stimuli presentation employed the same task timings as outlined above on both days. Crucially, we did not provide any feedback to participants during testing. Using only one auditory noise level per participant on the testing day allowed us to evaluate the effects of auditory benefit at different levels of visual evidence. We presented 720 trials—divided equally between all stimulus categories (i.e., face/car, V/AV, and four levels of visual evidence)—in short blocks of 60 trials with 60-second breaks between blocks. The entire task on the testing day lasted approximately 45 minutes. EEG data were collected only during the testing day.

**Behavioral analysis**. Our main behavioral analysis quantified participants' behavioral performance (i.e., decision accuracy and RTs) in the data collected during the testing day, using two separate GLMMs. GLMMs are superior to traditional repeated measures ANOVA analysis as their random effects structure better accounts for inter-participant variability, and allows for mixing of categorical and continuous variables[91]. Both models included all main effects and interactions of our two predictor variables, modality (V and AV) and visual evidence (27.5, 30, 32.5, and 35%), along with by-participant random slopes and random intercepts for the modality main effect. These random effects structure was justified by our design and adopted for reasons of parsimony. We employed post hoc likelihood-ratio ($\chi^2$) model comparisons to quantify the predictive power and significance of all main effects and interactions included in both GLMMs. These likelihood-ratio ($\chi^2$) model comparisons compared the full model (i.e., a model including all main effects, interactions, and random effects) to a reduced model, excluding the predictor or the set of predictors in question. Only results and statistics of the post hoc model comparisons are reported in the main results section. We performed these GLMM analyses using the lme4 package[92] in RStudio[93], specifying a binomial logit model in the family argument of the glmer function for decision accuracy, a binary dependent variable, and a gamma model for RT, a continuous dependent variable while selecting the bobyca optimizer. The predictor modality was entered in mean-centered form (deviation coding), whereas the predictor visual evidence (four levels) was entered using mean-centered backward difference coding. By using mean-centered coding schemes, we accounted for small imbalances in trial numbers between a predictor's levels. Random correlations were excluded for both GLMMs.

To quantify evidence for and against specific nonsignificant interaction effects in our two GLMMs, we complemented these models with model comparisons of two Bayesian linear mixed models (using the lmBF function and default priors of the BayesFactor package[94] in RStudio[93]). We report a Bayes Factor indicating the available evidence for the alternative model (i.e., a reduced model omitting the interaction in question) given the data and a proportional error estimate for the Bayes Factor resulting from 500.000 Markov chain Monte Carlo (MCMC) iterations. All models in this study used single-trial data as input and are based on the following mean amount of trials per condition across participants: $V_{car} = 178.53$, $V_{face} = 178.78$, $AV_{car} = 178.33$, and $AV_{face} = 177.88$. The respective mean absolute deviation was $V_{car} = 1.74$, $V_{face} = 1.6$, $AV_{car} = 2.18$, and $AV_{face} = 2.81$ trials. Note that 180 trials per condition were originally presented to all participants.

To quantify whether single-trial RT distributions are bimodal, we standardized (z-scored) RTs on the participant level before fitting a mixture of exponentially modified Gaussian (expGaussian) distributions (using maximum likelihood estimation) to the resulting RT distribution. Further, to formally rule out that our choice of participant-specific levels of auditory evidence could exclusively explain individual improvements in decision accuracy in AV trials, we correlated these measures across participants, using a robust bend correlation analysis[95]. Specifically, we evaluated whether the individual levels of auditory noise correlated with the difference in accuracy between V and AV trials (i.e., accuracy$_{AV}$–accuracy$_V$) across participants. As part of this correlation analysis, we computed the mean accuracy across all trials of each level of visual evidence and modality for each participant separately. In addition, to demonstrate that participants who performed well in V trials also performed well in AV trials, we complemented the above analysis by correlating decision accuracy (one value per participant calculated across visual coherence levels) between V and AV trials, using robust bend correlation analysis[95].

**EEG data acquisition and preprocessing**. We acquired continuous EEG data in a sound-attenuated and electrostatically shielded room from a 64-channel EEG amplifier system (BrainAmps MR-Plus, Brain Products GmbH, Germany) with Ag/AgCl scalp electrodes placed according to the international 10–20 system on an EasyCap (Brain Products GmbH, Germany). A chin electrode acted as ground and all channels were referenced to the left mastoid during recording. We adjusted the input impedance of all channels to <20 kΩ. The data were sampled at a rate of 1000 Hz and underwent online (hardware) filtering by a 0.0016–250 Hz analog band-pass filter. We used PsychoPy[90] and Brain Vision Recorder (BVR; version

1.10, Brain Products GmbH, Germany) to record trial-specific information, including experimental event codes and button responses simultaneously with the EEG data. These data were collected and stored for offline analysis in MATLAB. Offline data preprocessing included applying a software-based fourth-order butterworth band-pass filter with cutoff frequencies between 0.5 and 40 Hz. To avoid phase-related distortions, we applied these filters noncausally (using MATLAB filtfilt). Finally, the EEG data were re-referenced to the average of all channels.

We removed eye movement artifacts, such as blinks and saccades, using data from an eye movement calibration task completed by participants before the main task on the testing day. During this calibration task, participants were instructed to blink repeatedly upon the appearance of a black fixation cross on light gray background in the center of the screen before making several lateral and horizontal saccades according to the location of the fixation cross on the screen. Using principal component analysis, we identified linear EEG sensor weights associated with eye movement artifacts, which were then projected onto the broadband data from the main task and subtracted out[45]. We excluded all trials from all subsequent analyses where participants exceeded the RT limit of 1.5 s, indicated a response within <300 ms after onset of the stimulus or the EEG signal exceeded a maximum amplitude of 150 $\mu$V during the trial (0.8%, 0.06%, and 0.03% of all trials across participants, respectively).

**EEG data analysis**. We employed a linear multivariate single-trial discriminant analysis of stimulus- and response-locked EEG data[45,46] to identify early sensory and late decision-related EEG components discriminating between face and car trials as in previous work (e.g., refs. [36–38,42]). We performed this analysis separately for V and AV trials to independently identify the sensor signals discriminating the relevant visual evidence in each sensory modality condition, and allow direct comparisons between them in terms of overall discrimination performance. All single trials were included in all discriminant analyses.

Specifically, we identified a projection of the multichannel EEG signal, $x_i$, where $i = (1…N$ trials), within short time windows (i.e., a sliding window approach) that maximally discriminated between face and car trials (i.e., V discrimination: face-vs-car; AV discrimination: face/speech vs car/street sounds). All time windows had a width of 60 ms and onset intervals every 10 ms. These windows were centered on and shifted from −100 to 1000 ms relative to stimulus onset on stimulus-locked data and from −600 to 500 ms relative to the response button press on response-locked data. Specifically, a 64-channel spatial weighting $w(\tau)$ was learned by means of logistic regression[45] that achieved maximal discrimination within each time window, arriving at the one-dimensional projection $y_i(\tau)$, for each trial $i$ and a given window $\tau$:

$$y(\tau) = w(\tau)^T x(\tau) = \sum_{i=1}^{D} w_i(\tau) x_i(\tau).$$ (1)

Here, $T$ refers to the transpose operator and $D$ refers to the number of EEG sensors. In separating the two stimulus categories, the discriminator was designed to map component amplitudes $y_i(\tau)$ for face and car trials, to positive and negative values, respectively. These values are a weighted reflection of all available neural evidence with respect to the specific decision task (face-vs-car) that we asked participants to perform. By performing separate analyses for each modality condition, any unspecific effects present across trials—such as memory recollection or attention—would not be contributing to the estimation of the relevant classification weights separating face from car trials, and would effectively be subtracted out[96].

To quantify the performance of our discriminator for each time window, we used the area under a ROC curve[97], referred to as an $A_z$ value, combined with a leave-one-trial-out cross-validation procedure to control for overfitting[36–38,42]. Specifically, for every iteration, we used $N-1$ trials to estimate a spatial filter $w$, which was then applied to the left out trial to obtain out-of-sample discriminant component amplitudes ($y$) and compute the $A_z$ value. Moreover, we determined significance thresholds for the discriminator performance (rather than assuming an $A_z$ of 0.5 as chance performance) using a bootstrap analysis, whereby face and car labels were randomized and submitted to a separate leave-one-trial-out test. This randomization procedure was repeated 1000 times, producing a probability distribution for $A_z$, which we used as reference to estimate the $A_z$ value leading to a significance level of $p < 0.05$ (participant average $A_z\text{sig} = 0.57$). Note that this EEG analysis pipeline was performed on individual participants such that each participant became their own replication unit[98].

Finally, the linearity of our model allowed us to compute scalp projections of our discriminating components resulting from Eq. (1) by estimating a forward model as:

$$a(\tau) = \frac{x(\tau)y(\tau)}{y(\tau)^T y(\tau)},$$ (2)

where the EEG data ($x$) and discriminating components ($y$) are now in a matrix and vector notation, respectively, for convenience. Such forward models can be displayed as scalp topographies and interpreted, as the coupling between the observed EEG and the discriminating component amplitudes (i.e., vector $\alpha$ reflects the electrical coupling of the discriminating component $y$ that explains most of the activity in $x$). These forward models were computed separately for V and AV face-vs-car discriminant analyses.

**Optimizing number of distinct spatiotemporal components**. During periods of sustained significant discriminating activity, we used the forward model estimates resulting from Eq. (2) above to identify temporal transitions between different components based on differences in scalp distribution, which are typically suggestive of changes in the underlying cortical sources. Specifically, we used a $k$-means clustering algorithm using a Euclidean distance metric on the intensities of vector $a(\tau)$ for the entire time range of interest and optimized $k$ (i.e., the number of different time windows with similar scalp topographies) using silhouette values[99], as implemented in MATLAB's evalclusters function. Our results remained robust regardless of the choice of criterion (e.g., Silhouette, CalinskiHarabasz, etc.), the distance metric used for clustering, and the conditions it was applied to (i.e., V or AV trials). We used the resulting temporal components in all relevant EEG analyses.

**Temporal cluster-based bootstrap analysis**. To quantify if and when the discriminator performance differed between V and AV trials, we used a percentile bootstrap technique for comparing the group-level $A_z$ difference between two dependent samples[49]. Specifically, on a sample-by-sample basis, we created a distribution of shuffled $A_z$ difference scores (i.e., AV–V) across participants (drawing with replacement). We repeated this shuffling procedure 1000 times for each sample, whereby we created a random bootstrap distribution of median $A_z$ difference scores from every iteration. We computed the median of this bootstrap distribution for a given sample along with the 95% confidence interval (2.5–97.5%) of the resulting distribution of median difference scores. To test whether our bootstrapped median difference was significantly different from zero for each sample, we compared it against the lower bound of the estimated confidence interval (i.e., at the 2.5% threshold; $p < 0.025$).

To form contiguous temporal clusters and avoid transient effects due to false positives, we required a minimum temporal cluster size of at least three significant samples. This threshold was determined by means of the 95th percentile of a data-driven null distribution of maximum cluster sizes. Specifically, while in the analysis above the relationship between adjacent samples was preserved, here, we first applied a permutation procedure (i.e., shuffling temporal samples without replacement) to abolish the relationship across temporal samples, while keeping the relative difference between V and AV $A_z$ values unchanged, for each sample and participant. We generated the null distribution of maximum cluster sizes by computing and storing the maximum number of adjacent significant samples of the largest cluster for each of the 1000 iterations. Similar to the analysis on our original data, we performed this analysis on the discriminator performance ($A_z$) of both stimulus- and response-locked data (Figs. 3b and 4b, respectively), which yielded an average of at least three significant samples. This procedure corrects for multiple comparisons and is comparable to the temporal cluster-based nonparametric permutation test reported in ref. [100].

This entire procedure determined the extent of the temporal window used for the selection of the single-trial EEG component amplitudes ($y$-values), which we subsequently included in the neurally informed drift diffusion modelling analysis (see section below). Since our sample-based procedure was performed directly on discriminator accuracy ($A_z$), these times effectively represent the centers of the original discrimination windows, which consider data from a wider window (60 ms). To capture the full extent of these windows, we extended the selection window by 30 ms on either side of the significant clusters determined by our temporal cluster-based bootstrap analysis.

To ensure that neural effects were also reliably traceable in individual participants without group-level averages masking variability, we also computed the proportion of participants who demonstrated a participant-level effect in line with the general group-level effect per sample (that is, higher AV $A_z$ value for a given sample, see Figs. 3c and 4c). We performed these statistical analyses building on MATLAB code obtained from the Figshare and GitHub repositories associated with refs. [49,50].

To quantify evidence for and against the effects of sensory modality and decision accuracy on the subject-specific component amplitudes ($y$) based on trial accuracy (Figs. 3e and 4e), we computed two additional Bayesian linear mixed models analyses (using the generalTestBF function and default priors of the BayesFactor package[94] in RStudio[93]). Here, splitting trials into correct and incorrect responses, we report a Bayes Factor indicating the available evidence for the alternative model (i.e., a larger model including the predictor in question compared to a reduced model omitting the predictor in question) given the data. Note, when examining an interaction between sensory modality and decision accuracy the alternative model is the one omitting the interaction term.

Lastly, we performed a robust bend correlation analysis[95] to test the topographical consistency between the late stimulus-locked and response-locked components. We computed the average scalp map (i.e., forward models) across participants at the point of peak discrimination for the two components (500 ms poststimulus and 100 ms prestimulus, respectively) and assessed their similarity by computing their correlation. We also used two similar bend correlation analyses to test the extent to which the individual onset times in the Early component predicted (1) the difference in the stimulus-locked Late component discriminator amplitudes across the two modalities (AV vs V), and (2) the peak time of the stimulus-locked Late component.

**Hierarchical drift diffusion modelling of behavioral data**. We fit the participants' performance (i.e., face or car choice and RT) with an HDDM[101]. Similar to the traditional DDM, the HDDM assumes a stochastic accumulation of sensory evidence over time toward one of two decision boundaries representing the two choices (face or car). The model returns estimates of internal components of processing, such as the rate of evidence accumulation (drift rate), the distance between decision boundaries controlling the amount of evidence required for a decision (decision boundary), a possible bias toward one of the two choices (starting point) and the duration of nDT processes, which include stimulus encoding and response production.

The HDDM uses MCMC sampling to iteratively adjust the above parameters to maximize the summed log-likelihood of the predicted mean RT and accuracy. The DDM parameters were estimated in a hierarchical Bayesian framework, in which prior distributions of the model parameters were updated on the basis of the likelihood of the data given the model, to yield posterior distributions[52,101,102]. The use of Bayesian analysis, and specifically the HDDM, has several benefits relative to traditional DDM analysis. First and foremost, this framework supports the use of other variables as regressors of the model parameters to assess relations of the parameters with other physiological or behavioral data[51,76,78–80,103]. This property of the HDDM allowed us to establish the link between the EEG components and the aspects of the decision-making process they are implicated in. Second, posterior distributions directly convey the uncertainty associated with parameter estimates[102,104]. Third, the Bayesian hierarchical framework has been shown to be especially effective when the number of observations is low[105]. Fourth, within this hierarchical framework, all observers in a dataset are assumed to be drawn from a group, which yields more stable parameter estimates for individual participants[52]. To implement the hierarchical DDM, we used the Wiener module[101] in JAGS[106], via the Matjags interface in MATLAB to estimate posterior distributions. For each trial, the likelihood of accuracy and RT was assessed by providing the Wiener first-passage time distribution with the three model parameters (boundary separation, nDT, and drift rate). Parameters were drawn from group-level Gaussian distributions. The means and SDs of these group-level distributions had non-informative normally or uniformly distributed priors. Specifically, all SD priors were uniformly distributed $U(0.01, 2)$. The mean priors of nDT, boundary separation, and starting point were also uniformly distributed: nDT $\sim U(0.01, 1)$, $\alpha \sim U(0.01, 3)$, $\beta \sim U(0.1, 0.9)$. The priors of all the regression coefficients $\gamma_i$ means were Gaussians $N(0, 3)$. For each model, we ran five separate Markov chains with 5500 samples each; the first 500 were discarded (as "burn-in") and the rest were subsampled ("thinned") by a factor of 50 following the conventional approach to MCMC sampling, whereby initial samples are likely to be unreliable due to the selection of a random starting point, and neighboring samples are likely to be highly correlated[101]. The remaining samples constituted the probability distributions of each estimated parameter from which individual parameter estimates were computed.

To ensure convergence of the chains, we computed the Gelman–Rubin $\hat{R}$ statistic (which compares within-chain and between-chain variance), and verified that all group-level parameters had an $\hat{R}$ close to 1 and always lower than 1.03. For comparison between models, we used the DIC, a measure widely used for fit assessment and comparison of hierarchical models[54]. DIC selects the model that achieves the best trade-off between goodness-of-fit and model complexity. Lower DIC values favor models with the highest likelihood and least degrees of freedom.

We first estimated a nHDDM that used our EEG discrimination analysis to inform the fitting of the behavioral data. In this model, we input the single-trial RTs and (face or car) choices of all 40 participants, and hypothesized that the evidence accumulation rate during each trial would be dependent on the amount of neural evidence about face or car choice in that trial. Therefore, as part of the model fitting within the HDDM framework, we used the single-trial EEG measures of the face-vs-car discrimination analysis as regressors of the drift rate ($\delta$) as follows:

$$\delta = \gamma_0 + \gamma_1 * y_{\text{Early}}^s + \gamma_2 * y_{\text{Late}}^s * C, \qquad (3)$$

where $y_{\text{Early}}^s$ and $y_{\text{Late}}^s$ are the single-trial discriminator amplitudes of participant-specific stimulus-locked Early EEG components (individual peak $A_z$ across V and AV in the time range 180–360 ms poststimulus) and Late EEG components (individual peak $A_z$ difference between AV and V in the time range established in Fig. 3b; 490–540 ms (expanded further by 30 ms on either side to account for the resulting $A_z$ values being obtained with 60 ms training windows centered on these times)), respectively. The coefficients $\gamma_i$ weight the slope of the drift rate by the values of $y_{\text{Early}}^s$ and $y_{\text{Late}}^s$ of that specific trial, with an intercept $\gamma_0$. Here, we estimated $\gamma_i$'s for each participant and sensory condition. $C$ is the phase coherence level of the image presented in each trial. This value represents the quality of visual evidence available in each trial and has been shown to be proportional to the amplitude of the Late component[38,42,43]. Hence, by using these regression coefficients, we were able to test the influences of each of the two identified components on the drift rate in both sensory conditions[78]. Posterior probability densities of each regression coefficient were estimated using the sampling procedure described above. Significantly positive (negative) effects were determined when >99.9% of the posterior density was higher (lower) than 0. To test the significance of differences between the two sensory conditions (V vs AV), we performed a "hierarchical" $t$ test comparing the population-level distributions of

the parameters under consideration. This statistical testing has been shown to reduce biases induced by ignoring the hierarchical structure of the model (and testing at the participant level) and to actually yield conservative effect sizes[107].

For comparison, we also estimated a HDDM without including any neural correlates. We fit the HDDM to RT distributions for face and car choices conditioned on the sensory condition (V or AV) for each trial. Overall drift rate, boundary separation, starting point, and nDT were estimated for each individual participant and were dependent on the sensory condition. As per common practice, we assumed that evidence strength affected the drift rate; thus, we modeled a linear relationship between drift rate and coherence level[81].

**Reporting summary**. Further information on research design is available in the Nature Research Reporting Summary linked to this article.

## Data availability
The full neural and behavioral data required to reproduce the main analyses supporting this work, as well as the visual stimuli used in this study are available from the study's Open Science Framework repository (https://osf.io/rhx6y/). The raw EEG dataset is available from the corresponding authors upon request. Source data are provided with this paper.

## Code availability
Linear discriminant analysis code can be downloaded from the study's Open Science Framework repository (https://osf.io/rhx6y/). Code for reproducing all other analyses is available from the authors upon request.

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

## Acknowledgements

This work was supported by the Economic and Social Research Council (ESRC; grant ES/L012995/1 to M.G.P.), the European Commission (H2020-MSCA-IF-2018/845884, "NeuCoDe" to I.D.), the Physiological Society (2018 Research Grant Scheme to I.D.) and the European Research Council (ERC-2014-CoG/646657 to C.K.). G.D.S. was supported by an Engineering and Physical Sciences Research Council (EPSRC) doctoral training program.

## Author contributions

L.F., G.D.S., C.K., and M.G.P. designed the study. L.F. and G.D.S. performed the experiments. L.F., I.D., G.D.S., C.K., and M.G.P. analyzed the data and wrote the paper.

## Competing interests

The authors declare no competing interests.
