## [Peer Review File · Nature Communications]

Reviewers' Comments:

Reviewer #1:

Remarks to the Author:

In the current experiment De Sousa and colleagues use a very simple and elegant setup to delineate the contribution of early and late sensory processing to multisensory decision making. The manuscript is – with the exception of some formatting issues – very well structured and written, and the analytic approach is sound.

However, there are a number of major and minor issues to be resolved before I can recommend publication. My biggest concern relates to the use of visual inspection in identifying components of EEG activity used in later analyses without proper justification. This unclear description of especially the early component could severely influence the results and hence much of the interpretation and discussion of the results.

Please note that I comment on the methods section before going on with the results section and discussion.

Line 36ff: Please also see a recent review by Keil & Senkowski on a multistage model, which might be relevant for the discussion of which processing stages are involved in multisensory integration (Keil & Senkowski, 2018). This is also true for the discussion in line 352.

Line 456: The Internet is a rather ambiguous source for the car images used in the experiment. Could you be more precise with respect to the description of the images, their sources and whether you obtained permission to use the images in research from the copyright holders? The same is true for line 469.

Line 465: For how long did you present the visual stimuli?

Line 513 and throughout the text: I suggest the use of “participant” instead of subject, as you did in the description of the sample, to highlight the voluntary participation (i.e. participants are not subject to a treatment).

Line 539: How did you compute the response times over trials? Are these means or medians? In figure 2, you mention “averages”. However, RTs are usually not normally distributed, and due to the skewed distribution, the mean is not appropriate (see figure 5b). I suggest using the median instead.

Line 580: Please provide more details on the offline-filters (type, order, transition bands).

Line 656: In the cluster-based correction for multiple comparisons by Maris & Oostenveld, the critical cluster size is based on the data. How did you come up with the threshold of three samples in the current analysis?

Line 693f: The citations are formatted incorrectly here. Please double check.

Line 721: Please remove the dots between “face vs. car”

Line 726 & 728: How did you come up with these time intervals? The first doesn't correspond to the intervals identified in the stimulus-locked analysis.

Line 116: Please be careful with interpreting non-significant results. Could you quantify this apparent

enhancement (e.g. using Bayes Factors)?

Line 135: Multisensory redundant information usually speeds up response times (Miller, 1982). Related to my comment to line 539, this contrary result might be due to the use of the mean instead of the median, thereby giving undue weighting to outliers.

Line 158: Related to my comment to line 726, could you describe in more detail how you identified the two temporally specific components?

Figure 3: Related to the previous point, I don't follow your argument of two temporally specific components. First, the small inserts with the topographies don't illustrate these components. As a side-note, please add a color bar and indicator what is depicted here. Second, "EarlyS" and "LateS" are floating around the line graph without a clear indicator to where they belong. I suggest a clearer marking of the components. Third, please name the colors corresponding to the conditions in Figure 3a directly, not just in the figure legend. Finally, please clearly label the small bar-plot inset. If this belongs to the comparison illustrated in 3b, then I suggest moving it there.

Line 180: The citation to Rousselet 2017 is not formatted correctly. Please double check. Also, I'm not sure this citation is at the right place, as it does not deal with cluster-based permutation analyses, but with group comparisons using bootstrapping techniques (i.e. move the reference down two sentences).

Line 209: Please be careful with visual inspection of data and drawing conclusions based on these inspections. Could you formally compare the topographies, e.g. using the topographic dissimilarity measure (Murray, Brunet, & Michel, 2008)?

Figure 4: As with figure 3, I suggest clarifying the relationship between the topographies (which again lack a color bar), the identified peaks, the line plots and the inset with the bar-plots.

Line 268: Related to my comments to line 158 and 726, how did you come up with these intervals? From what I gathered from the previous description of the stimulus-related analysis, you identified two peaks by visual inspection (around 220/230 ms and 460/500 ms), and found a significant Az-difference between 490 and 540 ms. The interval mentioned in line 726 (170 to 250 ms) is not further motivated, but used here for further analyses.

Line 286: Again, related to the previous point, this is not really surprising, given that in the (as far as I can tell) arbitrary "early" interval, no differences in stimulus discrimination accuracy have been found.

Line 353: The citation of Bizley et al. is not formatted correctly.

References:

Keil, J., & Senkowski, D. (2018). Neural Oscillations Orchestrate Multisensory Processing. *The Neuroscientist*, 24(6), 609–626. <http://doi.org/10.1177/1073858418755352>

Miller, J. (1982). Divided attention: evidence for coactivation with redundant signals. *Cognitive Psychology*, 14(2), 247–279.

Murray, M. M., Brunet, D., & Michel, C. M. (2008). Topographic ERP Analyses: A Step-by-Step Tutorial Review. *Brain Topography*, 20(4), 249–264. <http://doi.org/10.1007/s10548-008-0054-5>

Reviewed by Julian Keil

Reviewer #2:

Remarks to the Author:

In this paper De Sousa and colleagues seek to address a major question regarding whether perceptual improvement during multisensory decision is best explained by so-called 'early' sensory processing or 'late' post-sensory change in decision making dynamics. To that aim, the authors employed a visual object categorization task while participants were exposed to either visual-only stimuli or to audio-visual stimuli. They analyzed concurrently recorded EEG data using discrimination analysis to distinguish object categories. Based on this method, the authors selected two EEG components peaking at different latencies: an Early and a Late component. Comparing the two conditions, they found that complimentary auditory information in audio-visual context amplified the Late but not the Early component. Last, they used neurally-informed drift diffusion model and showed that performance gain in multisensory context arose from enhanced single-trial representation of the Late component.

This study is timely and its purpose relevant for the community as it bridges two major fields in neuroscience that are multisensory integration and decision making. The methods used for the analysis are elaborated and appropriate. Nonetheless, several major concerns prevent to endorse the conclusion of the study. Which includes the interrogation whether genuine multisensory integration process is at play during the task. I very much hope that the comments outlined below will be helpful to alleviate some of the issues.

1) The analysis of behavioral data reveals that in this study the multisensory gain comes with slower response times. This result is intriguing as it is not concordant with typical multisensory effects/benefits where faster response times parallel better accuracy. Such unexpected finding raises some interrogations that are not rigorously considered in the manuscript.

Here, an extreme hypothesis can be envisaged: the multisensory context used in this experiment did not lead to multisensory integration per se. For instance because the auditory stimulus is too short – 50ms minus the 10ms on/off ramp – to carry the semantic information that must be integrated with visual information. Under this premise, we could conjecture that participants based their decision either on visual information only or on auditory information only (while the stimuli were short, this later case was made possible because participants learnt during a training session to associate the auditory snippets with the two categories). That is, performance in the multisensory condition increased simply because the complementary auditory information was used whenever visual information did not permit to categorize the stimulus. It followed that when participant based their decision on auditory information response time increased. This increase being caused by modality switch cost in resource deployment and/or by further auditory processing. One way to explore this possibility would be to compare normalized RT distributions (e.g. z-scored by participant) between conditions to assess if the multisensory context leads to a bimodal RT distribution. Another common practice would be to complement the analyses with EEG data recorded for auditory-only trials. Based on their modeling approach, the authors postulated that longer RTs in the multisensory context reflected additional encoding time needed when auditory processing was at play. However this does not seem to be supported by the electrophysiological results: the discrimination analysis depicted in Figure 3 indicates that in the multisensory condition the Early component peaks earlier. It is only after 300 ms that we can see a sort of slow down (or an acceleration in the visual-alone condition), leading to a delayed Late component in the multisensory condition as to compare to the visual condition. This change in the dynamic is probably at the origin of the slower RTs in the multisensory context. It would be important to further analyze the variation in the processing dynamic to evaluate acceleration/slow

down between conditions.

To rule out that the higher recognition performance in audiovisual context was not exclusively related to complimentary auditory information, the authors used a robust bend correlation analysis between subject-specific levels of auditory evidence with individual improvements in decision accuracy on audiovisual trials. This approach might be somehow too indirect as subject-specific auditory noise was estimated to support perithreshold performance. It could be more straightforward to directly correlate decision accuracy between both conditions (auditory and audio-visual). Still the conducted analysis revealed that the amount of auditory evidence explained a fraction of the variance in the accuracy across subjects. And, while the R-value from the correlation analysis is provided, the p-value is not indicated.

Another argument supporting the possibility that no multisensory integration is at play in this study relates to the absence of inverse effectiveness effect. The principle of inverse effectiveness in multisensory integration states that, as the responsiveness to individual sensory stimuli decreases, the strength of multisensory integration increases. Accordingly, one may assume that multisensory benefit would be linked to the level of visual evidence. This was tested and, unfortunately, no significant interaction was found between modality and the level of visual evidence.

2) The main claim of the paper relies on the discriminator performance being found significantly higher in the multisensory condition only at the latest time period, considered by the authors as post-sensory decision related processing. This is summarized at the beginning of the discussion where the authors wrote: "... AV trials were accompanied primarily by enhancements in a Late EEG component indexing decision-related processes. In contrast, an earlier EEG component encoding sensory (visual) evidence remained unaffected by the addition of complementary auditory evidence". These statements are somewhat hasty considering the results presented in the paper.

First, regarding the Early and Late components, the authors postulate that the components reflect two functional steps: encoding sensory and post-sensory visual evidence, respectively. However this is not demonstrated and only assumed based on the look of component topographies. Considering the previous studies conducted by some of the authors in the visual domain, additional analysis can be conducted to prove their point, especially in the audio-visual context. The authors may also want to think through the work from other groups who did investigate the link between sensory encoding and decision formation (for instance Romo and colleagues in primates, or O'Connell and colleagues in humans). Moreover, the discriminant analysis actually revealed three peaks but only the first and the last one were considered. What was the rationale to leave aside the second and highest peak? Second, the lack of statistical significance does not permit to reach a final conclusion about the absence of an effect. It could be the case that the relevant signal is too small in the present study to lead to significant difference, or that the effect is only present in a subset of the participants. In Figure 3 part b, the difference between discriminator performance for visual and audio-visual conditions clearly show two peaks at 200 and 300 ms almost reaching statistical significance. This is further confirmed in Figure 3 part c, where we can see that the two peaks at 200 and 300 ms are shared by at least 60% of the participants. These two peaks are also visible in the response-lock analysis presented in Figure 4. Also, it would be interesting to investigate if there are two subsets of participants: the early and the late 'integrators'.

Minors:

- Several information concerning the auditory stimuli are lacking. For instance, the normalization procedure for the pictures is described in details but not the procedure to normalize the sounds. Also, the amount of sounds per categories is not indicated.
- Can the authors specify the mean amount of trials per condition (plus-minus mean absolute deviation)?
- In the description of the linear multivariate single-trial discriminant analysis, it is not mentioned if only correct trials were used in the procedure.

- Spatial correlation was computed between the components isolated in the visual-only condition and the audiovisual condition, but this is not detailed in the material and method.
- In Figure 3 and 4, it would be pertinent to extend the x-axis to cover the entire period of interest. That is to include in both figures (and to indicate graphically) the onset of the stimulus and the mean response time plus some baseline on each side to ensure that the discrimination analysis goes back to chance level. Also the color scale for the topographies is missing.
- Figure 5 part a, the graphical representation of the HDDM parameters is confusing. It depicts the estimates of the variance (σ) for τ and α , but this is not mentioned in the text. Moreover, μ and σ for τ point toward α , while μ and σ for α point toward τ .
- Figure 5 part b, RT distributions are substantially different between Car-choice and Face-choice. Is it related to difference in performance between the two categories? Can the authors provide performance and mean RTs for the two categories (+/- m.a.d.)?
- At the end of the neurally-informed modelling analysis, the authors underlined the fact that "the average difference in RTs (37ms) is very similar as the average non-decision difference between the two conditions (38ms)". Yet, did the authors tried to correlate the two variables across participants?
- In the report of the HDDM constrained only by the behavioral data, the drift rate of evidence accumulation was not provided. Can the authors add this information?

Reviewer #3:

Remarks to the Author:

In their paper "Auditory information enhances post-sensory, but not early sensory, visual evidence during multisensory decision making", De Sousa and colleagues perform an EEG study of multisensory perceptual choice and combine this with cognitive modeling to investigate how and when the benefit of multisensory information impacts on the choice process. Using an EEG discriminability analysis established by the senior author, they identify an early and a late EEG component that allow to separate visual evidence for the two choice options (face vs. car). Only the late component distinguishes between the multisensory and the purely visual conditions in the task, both stimulus- and response-locked. The authors go on and combine the EEG data with hierarchical modeling of the drift diffusion model (DDM) and find that only the weight of the late EEG component on the drift rate is stronger in the multisensory condition. Furthermore, they show that fitting the DDM without neural data could lead to different (and possibly wrong) conclusions (i.e., a higher boundary separation in the multisensory condition).

Overall, the paper is written in a very clear, succinct and precise style, and the study addresses a timely research question (i.e., the computational and neural principles of multisensory perceptual decision making). The related literature is introduced and discussed sufficiently, the methods and results are presented clearly, and the potential impact of the current study is put into context adequately. Given that there have already been some publications on modeling multisensory integration with the DDM (e.g., Drugowitsch et al., 2014, eLife) and on EEG patterns of multisensory integration, the novelty of the current work lies in the sophisticated combination of EEG with cognitive modeling. I agree with the authors that this provides potentially interesting and important new conclusions, in particular because the modeling results seem to differ substantially depending on whether the neural data is included or not. However, I have several remarks on the manuscript, most of them of methodological nature, that need to be addressed to ensure that the modeling (but also the EEG) conclusions are indeed justified.

Major points:

1. Cognitive modeling: When estimating the DDM, the authors seem to ignore a potential starting

point bias. In the methods, they state that the task would not induce such a bias and refer to previous work. However, Figure 5b seems to indicate that car choices were more frequent than face choices, which could speak for such a starting point bias effect. I might be wrong here (descriptive statistics about the choice frequencies would be helpful), but if not, I think this possibility should be accounted for by treating the starting point as a free parameter in the DDM.

Second, I was surprised that the different visual conditions were fit independently from each other rather than assuming a "linking" function (e.g., linear, power, logistic) that connects phase coherence with the drift rate (like in Philiastides et al., 2014, *J Neurosci*). First, this should just be more efficient. Second, the current approach apparently yields a beta weight of the EEG data on the drift rate for each coherence level (see methods). How do you integrate those then within a participant? Taking the mean across coherence levels? Furthermore, in the methods it says that statistical inference on the beta weights were made by looking at the posterior density (i.e., in a Bayesian way), but frequentist statistics are reported in the results (page 14). In my view, the best approach would be to assume that single-subject beta weights are drawn from (normally distributed) group-level distributions for means and SDs, and then to perform the statistical tests on those group-level means (see Boehm et al., 2018, *Behav Res Method*).

Third, a more precise specification of the priors is required.

2. EEG analysis: I am wondering whether the fact that the audiovisual and the visual conditions induced differently long non-decision times, and in particular differently long sensory encoding times, could pose a conceptual problem for the EEG analysis. The early and late components are compared to each other in a stimulus-locked way at every point in time, but if the evidence accumulation process starts later in the audiovisual condition, then the question is whether comparing the same time points across conditions actually makes sense. Obviously, this concern does not affect the response-locked analyses.

I am further wondering whether the statistical requirement of having 3 consecutive time points at which the two conditions must differ at $p < .05$ is really a proper way of correcting for multiple comparisons. Is this just a rule-of-thumb (like $p < .001$ and 10 contiguous voxels as seen in many fMRI studies) or is it based on a more principled way of correcting for multiple comparisons (like an FWE- or FDR-correction in fMRI)? Perhaps the authors can elaborate on this.

3. Although they cite the study by Drugowitsch et al., 2014, *eLife*, I think this study and its comparison to the current results deserve a more thorough discussion. The behavioral findings are very different: in the Drugowitsch paper, participants make faster and sometimes even worse decisions in the multisensory case; in the current case, they are slower and more accurate. To me, this appears as if participants used different strategic approaches in the two studies. Why could that be the case? And is this interpretation in line with the current modeling results (which seem to speak against a threshold-adaptation account)?

Minor points:

4. Statistical analysis of the effects shown in Fig. 5c: I think it is more appropriate to test for effects of sensory conditions, early/late components and their interaction using an ANOVA rather than multiple paired t-tests.

5. How was the sample size determined?

6. Typos: line 177 ("as" rather than "a"); line 315 ("similar to" rather than "similar as"); line 327 ("a" rather than "as"); line 359 (delete the comma after "specifically"); line 721 ("face.vs.car").

Signed,
Sebastian Gluth

We thank the reviewers for their very constructive feedback. We have aimed to address all their comments and concerns in the revised manuscript, as detailed in the point-by-point replies below (see sections in blue font). Sections written in italicised Times New Roman font are direct quotes from the revised manuscript.

Reviewers' comments:

Reviewer #1 (Remarks to the Author):

In the current experiment De Sousa and colleagues use a very simple and elegant setup to delineate the contribution of early and late sensory processing to multisensory decision making. The manuscript is – with the exception of some formatting issues – very well structured and written, and the analytic approach is sound.

However, there are a number of major and minor issues to be resolved before I can recommend publication. My biggest concern relates to the use of visual inspection in identifying components of EEG activity used in later analyses without proper justification. This unclear description of especially the early component could severely influence the results and hence much of the interpretation and discussion of the results.

We would like to thank the reviewer for highlighting the need to better motivate the selection procedure for the two EEG components identified in this work.

In the original analysis, we initially defined broad temporal windows corresponding to the Early and Late components based on an extensive body of previous work, which we have replicated on a number of occasions. We subsequently selected individual-participant Early/Late components based on the actual peak A_z (classification) values obtained from every participant in each of these two windows (rather than purely by visual inspection).

Nonetheless, we very much appreciate the need to derive these components from the current data (rather than relying on “older” definitions of the temporal properties of these components). Motivated by the reviewer’s suggestion we ran an even more rigorous procedure for selecting these components, without any a priori assumptions on the number and timing of these components.

Specifically, to identify the number of relevant components in the entire range of significant classification as seen in Fig. 3a (180-600ms post-stimulus), we employed a temporal clustering approach based on differences in scalp map distributions, which are typically suggestive of changes in the underlying cortical sources/processes (see Materials and Methods on page 33f. for more details). More specifically, we used a k-means clustering algorithm and optimized k (i.e., the number of different time windows with similar scalp topographies). This procedure revealed the presence of two temporally distinct scalp representations with a transition point at 380ms post-stimulus, for both V and AV conditions. These spatial representations are consistent with our previously reported Early and Late components (Piliastides et al., 2006; Piliastides & Sajda, 2006a; Piliastides & Sajda, 2006b; Piliastides & Sajda, 2007; Ratcliff et al., 2009, Blank et al., 2013, Diaz et al., 2017, etc.), with centrofrontal and bilateral occipitotemporal activations for the Early and a prominent centroparietal activation cluster for the Late component (Fig. 3a, top). Our results

revealed the same two temporally distinct clusters regardless of the choice of the evaluation criterion and the distance metric used for clustering.

Finally, we extracted participant-specific component latencies – for each modality condition separately – by identifying the time points leading to peak A_z (classification) performance within each of the two windows identified by the clustering procedure. We used these definitions for all subsequent analyses. Hence, all analyses in the revised manuscript are based on objectively identified data-driven neural components. These components are consistent with what we would expect from previous work. We now describe this new approach in the Methods (pg. 33f.) and amended the Results accordingly (pg. 9f.).

Please note that I comment on the methods section before going on with the results section and discussion.

Line 36ff: Please also see a recent review by Keil & Senkowski on a multistage model, which might be relevant for the discussion of which processing stages are involved in multisensory integration (Keil & Senkowski, 2018). This is also true for the discussion in line 352.

Thank you for bringing this review paper to our attention. We now reference this review in our discussion (pg. 20 line 407ff.) in relation to possible multisensory system interactions in the brain, which we then link to our own findings as follows: *The processing of multisensory information requires the coordination of multiple mechanisms serving bottom-up, top-down and predictive coding processing (Keil & Senkowski, 2018). The neural implementation of these mechanisms involves a distributed neural network including primary sensory, parietal and frontal brain areas that interact with each other to form and shape multisensory perception (Cao, Summerfield, Park, Giordano, & Kayser, 2019; Rohe, Ehrlis, & Noppeney, 2019).*

Line 456: The Internet is a rather ambiguous source for the car images used in the experiment. Could you be more precise with respect to the description of the images, their sources and whether you obtained permission to use the images in research from the copyright holders? The same is true for line 469.

The car images employed in this experiment were obtained from multiple sources on the internet back in 2003, as no formal car database existed at the time. We ensured that no copyright restrictions were in place on the selected images, and modifications to the images were allowed. Subsequently, we modified the images by removing the background and placing all images on a uniform grey background. This car dataset is openly available to the community (it has been so, for a number of years now) and is available for download from our lab website:

<https://www.mphiliastides.org/en/resources>

The same holds true for the auditory sounds we used herein (no restrictions associated with their use or modifications were in place at the time of retrieval). To clarify these points, we

added additional text in the manuscript under the methods section in lines 536f. and 551f. on pages 25 and 26.

Line 465: For how long did you present the visual stimuli?

The presentation duration of the visual (and auditory) stimuli was 50ms. This information was originally presented in the legend of figure 1 and in the Methods section. For further clarity we now present this information early on in the Results section of the revised manuscript text itself (line 98 on page 6) and added another instance when describing the visual stimuli in the Methods section in line 547 on page 26.

Line 513 and throughout the text: I suggest the use of “participant” instead of subject, as you did in the description of the sample, to highlight the voluntary participation (i.e. participants are not subject to a treatment).

All instances of the word “subject” have been replaced by the word “participant” to highlight the voluntary aspect of taking part in this study.

Line 539: How did you compute the response times over trials? Are these means or medians? In figure 2, you mention “averages”. However, RTs are usually not normally distributed, and due to the skewed distribution, the mean is not appropriate (see figure 5b). I suggest using the median instead.

We would like to thank the reviewer for this thoughtful suggestion. To avoid introducing any potential biases due to skewed distributions, we have recomputed the RT data originally displayed in figure 2c and 2d to showcase the group median (see below, now figures 2d and 2e), rather than the mean as in the earlier version of the manuscript. We note that the main analysis of the RT effects remained unaffected, since the generalised linear mixed effects model we employed uses single-trial RT data and a gamma function that accommodates skewed distributions such as the ones commonly captured in RTs. The random effects structure we employed accommodates variation by participant, which further accommodates inter-individual differences in RTs. Therefore, the results of this model in the main text remained unchanged (p. 7f.).

Line 580: Please provide more details on the offline-filters (type, order, transition bands).

We apologise for this omission. We have applied a 4th-order butterworth bandpass filter with cutoff frequencies between 0.5-40 Hz. This information was added to the Methods section in line 684f., on pg. 31.

Line 656: In the cluster-based correction for multiple comparisons by Maris & Oostenveld, the critical cluster size is based on the data. How did you come up with the threshold of three samples in the current analysis?

We would like to thank the reviewer for this useful clarification request. We have now introduced a formal procedure to derive a cluster-based correction directly from our data (in the spirit of the Maris & Oostenveld paper). Specifically, we computed multiple null distributions of our A_z data by first applying a permutation procedure (i.e., shuffling temporal samples without replacement) to all temporal samples on the individual participant level, whereby we abolished the relationship between temporal samples. Importantly, by only shuffling the temporal order of the samples but not the difference scores between our modality conditions for each sample, the relative difference between unisensory (V) and multisensory (AV) A_z values remained unchanged for each sample and participant. This permuted data served as input to a subsequent sample-based bootstrapping procedure, which was identical to the bootstrapping procedure applied to obtain the median difference scores in our original analysis. Using this procedure, we computed the maximum number of adjacent significant samples of the largest cluster for each iteration and stored this value to build a null distribution of maximum cluster sizes. We repeated these steps 1000 times and selected the value corresponding to the 95th percentile of this distribution as minimum cluster size threshold for comparisons with clusters of significant samples in the original data. This data-driven, cluster-based correction for multiple comparisons yielded an average minimum cluster size of three consecutive samples for stimulus- and response-locked data quantifying the difference between V and AV A_z traces. More details of this procedure are now given in the Methods section on page 34f. (line 773ff.).

Line 693f: The citations are formatted incorrectly here. Please double check.

Thank you for spotting this discrepancy. The two APA style citations have been changed to their corresponding citation numbers (line 833 on pg. 37).

Line 721: Please remove the dots between “face vs. car”

The dots have been removed as requested in all relevant instances. Further, we changed all instances in which “face vs car” is part of a descriptive term to “face-vs-car”.

Line 726 & 728: How did you come up with these time intervals? The first doesn't correspond to the intervals identified in the stimulus-locked analysis.

Please see our first response to the reviewer's concern about the selection of the relevant EEG components. In short, our primary aim in this work was to identify time windows over which there was a significant difference in our ability to discriminate between face versus car

images across the V and AV conditions. In the original analysis due to the absence of significant differences between V and AV A_z traces until later on in the time course (after 490ms post-stimulus onset), we defined the temporal interval of the Early component based on our previous characterisation of this component in a unisensory (i.e., visual) context (Philiastides et al., 2006; Philiastides & Sajda, 2006a; Philiastides & Sajda, 2006b; Philiastides & Sajda, 2007; Ratcliff et al., 2009, Blank et al., 2013, Diaz et al., 2017, etc.).

On reflection, and taking the reviewer's suggestion under advisement, we have now introduced a more formal procedure for selecting these components purely based on the data of this audiovisual experiment and without any a priori assumptions on the number and timing of these components (details are provided in our earlier comment and in the Methods section). In doing so, we now initially identify the Early and Late components using the clustering procedure described above, using purely the outcome of our face-vs-car discrimination (i.e. discriminant scalp topographies) in each of the V and AV conditions. Only then, we go on to compare differences in A_z between the V and AV conditions.

The new interval definitions of the Early and Late components resulting from our temporal clustering procedure using the forward models (scalp maps) of the discrimination procedure are identical across the V and AV conditions and are as follows: 180ms to 360ms and 400ms to 600ms post-stimulus, respectively (line 183ff. on page 9f.). From within these intervals we extract participant-specific peak A_z latencies and show that there are no significant differences in the timing of the two components in the V and AV conditions (pg. 10).

Finally, using our modified bootstrapping procedure on single-participant A_z traces (see comment above), we find significant differences in the overall discrimination performance between the V and AV conditions in the time interval 490 – 540ms post-stimulus, which falls squarely within the temporal range of our Late EEG component (line 234ff., pg. 12).

Line 116: Please be careful with interpreting non-significant results. Could you quantify this apparent enhancement (e.g. using Bayes Factors)?

We agree with the reviewer(s) that quantifying the evidence for the null hypothesis over the alternative hypothesis using a Bayes Factor analysis would be the best way to provide evidence for the null. Our intention in the original manuscript was merely to provide a qualitative impression of the seemingly wider gap between V and AV accuracy, visible in the group averages (Fig. 2a). We included this statement in the former version of the manuscript for reasons of transparency. However, we acknowledge that these statements could have been misleading. Although we did not interpret these null effects further in the remainder of the manuscript, we decided to remove this part of the sentence on the appearance of statistically non-significant behavioural interaction effects (formerly line 116) to ensure clarity and reflect the uncertainty surrounding the interpretation of said null interaction effects (line 118ff., pg. 7).

In the revised behavioural results section, we now explicitly report the exact statistics for all likelihood-ratio (χ^2) model comparisons of all interactions between the predictor modality and the three separate predictors for visual coherence. Hence, we would like to offer the reader the opportunity to judge the evidence by themselves (modality x 27.5/30% coherence: $\chi^2 =$

0.60, $df = 1$, $p = .4376$; modality x 30/32.5% coherence: $\chi^2 = 0.01$, $df = 1$, $p = .9142$; modality x 32.5/35% coherence: $\chi^2 = 0.69$, $df = 1$, $p = .4047$). In addition, we complemented our linear mixed effects model approach with two Bayesian models (using the *lmBF* function of the *BayesFactor* package in RStudio) predicting accuracy and RT separately. One model included the predictors modality and visual coherence and their interactions (null model), whereas the other model included the same two predictors but without any interactions (alternative model). Both models also included a random factor by participant. Similar to the mixed effects model analysis, single-trial data served as input data for this Bayesian analysis. A direct post-hoc comparison of both models illustrated that there was very strong evidence for the “alternative model” without interactions given the data ($BF_{10} = 5030.05 \pm 0.62\%$), corroborating the absence of interaction effects (pg. 7).

The revised manuscript provides the same level of detail for the RT data reporting model comparisons for all interaction effects ($\chi^2 = 1.43$, $df = 1$, $p = .2322$; $\chi^2 = 1.53$, $df = 1$, $p = .2156$; $\chi^2 = .004$, $df = 1$, $p = .9522$, respectively) and very strong evidence for the alternative model without interactions given our data from the Bayesian analysis ($BF_{10} = 1848.15 \pm 0.53\%$).

Line 135: Multisensory redundant information usually speeds up response times (Miller, 1982). Related to my comment to line 539, this contrary result might be due to the use of the mean instead of the median, thereby giving undue weighting to outliers.

We agree with the reviewer that the mean could be susceptible to outliers. Therefore, as requested, we now only use the median for displaying and reporting response times. The results showed a slightly smaller overall increase in RTs across visual coherence levels for the median (*Median* = 29.1ms) as compared to the mean (*Mean* = 33.1ms). In either case, this increase in RTs is small and is illustrated by the revised figures 2d and 2e (also shown above as part of our response to an earlier comment) depicting group medians and their overlapping standard errors for all four visual coherence levels.

Miller’s (1982) race model does not apply to the data of our study because it assumes similar sensory encoding for the stimuli presented to both modalities and is centred around the idea of complete redundancy of the multisensory information. For this reason, it is normally applied only in contexts where the task-relevant nature of the stimulus is immediately apparent; e.g. in simple detection tasks and not in discrimination tasks as used here. Hence, in our view it is not that surprising that the data violate the race model assumptions. The increased reaction times likely stem from the auditory information being context dependent and complementary to the visual information rather than redundant. In other words, the slightly longer reaction times may indicate that the auditory sounds are not treated as “redundant” but rather as “complementary” pieces of information that require additional processing resources early on.

The nature of these slower response times could also be justified further by considering evidence from our neurally informed drift diffusion model. Our model suggests that non-decision times in the AV trials are longer compared to the V trials (by 38ms on average), which points to a prolonged initial encoding and processing of the complementary sensory evidence (possibly due to the auditory stream requiring slightly longer processing times). As highlighted above, this longer stimulus encoding duration may derive from the particular construction of the multisensory context in our task, where decisions are based on a broader

audiovisual context and a higher-level conceptualisation of the evidence, such as the presence of car-like objects and general sounds signalling a busy street (rather than fully congruent and hence complementary AV evidence). These considerations are discussed in more detail in the Discussion (pg. 22).

Line 158: Related to my comment to line 726, could you describe in more detail how you identified the two temporally specific components?

Please see our detailed responses to the comment regarding line 726 and our opening remarks about the selection of the Early and Late components above.

Figure 3: Related to the previous point, I don't follow your argument of two temporally specific components. First, the small inserts with the topographies don't illustrate these components. As a side-note, please add a color bar and indicator what is depicted here. Second, "EarlyS" and "LateS" are floating around the line graph without a clear indicator to where they belong. I suggest a clearer marking of the components. Third, please name the colors corresponding to the conditions in Figure 3a directly, not just in the figure legend. Finally, please clearly label the small bar-plot inset. If this belongs to the comparison illustrated in 3b, then I suggest moving it there.

Thank you for your suggestions on how to improve the visualisation of figure 3 (pg. 11). In short, following your suggestions we have 1) clearly marked the two components as they have been revealed by the new temporal clustering procedure (see previous responses) using shaded areas in the background, 2) added a wider range of scalp topographies to better capture the temporal evolution of these two components and their transition point using a vertical line (~380ms post-stimulus onset) along with a labelled colorbar, and 3) added an additional panel (d) depicting the former inset separately as it displays mean amplitude differences of the single-trial discriminator output values (Y_s rather than A_2) between V and AV of the significant time period during the Late component shown in panel b (marked by the solid black line along the x-axis).

Line 180: The citation to Rousselet 2017 is not formatted correctly. Please double check. Also, I'm not sure this citation is at the right place, as it does not deal with cluster-based permutation analyses, but with group comparisons using bootstrapping techniques (i.e. move the reference down two sentences).

We would like to thank the reviewer for this suggestion. In the process of making general modifications to this paragraph, we moved this citation further down to the end of the sentence describing the employed bootstrapping approach. It has now been moved to line 230 on page 12.

Line 209: Please be careful with visual inspection of data and drawing conclusions based on these inspections. Could you formally compare the topographies, e.g. using the topographic dissimilarity measure (Murray, Brunet, & Michel, 2008)?

We thank the reviewer for this suggestion. We have now formally compared the topographies using as similarity measure a robust correlation coefficient. We found a high similarity between the group average stimulus-locked Late component and the group

average response-locked component ($r_{bend}(38) = .88$ for V and $.86$ for AV). This correlation is suggestive of largely overlapping generators across these two components identified in each of the stimulus- and response-locked analyses. We included this information in line 257 on page 13 of the revised manuscript.

Figure 4: As with figure 3, I suggest clarifying the relationship between the topographies (which again lack a color bar), the identified peaks, the line plots and the inset with the bar-plots.

In correspondence with our changes to figure 3, we have implemented the identical changes in figure 4 (pg. 13), as both figures follow the same logic and only differ in the time point these data are locked to (i.e., stimulus- versus response-locking).

Line 268: Related to my comments to line 158 and 726, how did you come up with these intervals? From what I gathered from the previous description of the stimulus-related analysis, you identified two peaks by visual inspection (around 220/230 ms and 460/500 ms), and found a significant Az-difference between 490 and 540 ms. The interval mentioned in line 726 (170 to 250 ms) is not further motivated, but used here for further analyses.

As previously mentioned, we are now using a principled approach for defining these intervals exclusively derived from the current data, without any a priori assumptions from earlier work. We have discussed this approach in great detail in our earlier response to the reviewer above (please see our opening comment and our response to the comment regarding “Lines 726 & 728”) and in the revised manuscript (pg. 9f. and 33f.).

Line 286: Again, related to the previous point, this is not really surprising, given that in the (as far as I can tell) arbitrary “early” interval, no differences in stimulus discrimination accuracy have been found.

Indeed, we did not find any difference in discrimination accuracy between V and AV in the Early window (even after selecting all components with the rigorous procedure described above). We now include the Early component amplitudes originating from a wider time range in the model, which allows for directly comparing the contribution of each of these components (i.e., Early and Late) to drift rate (regardless of modality). We validate earlier work showing that both components are predictors of the rate of evidence accumulation. However, only the modulation of the Late component is shown to scale with the reliability of the available evidence (coherence levels here), thus reflecting the amount of evidence entering the decision process and ultimately the accuracy of the perceptual choice. We now included this information on page 17 of the revised manuscript.

Line 353: The citation of Bizley et al. is not formatted correctly.

We would like to thank the reviewer for spotting this issue. The APA citation has been removed, so that only the Nature numbering style citation remains (line 408, pg. 20).

References:

Keil, J., & Senkowski, D. (2018). Neural Oscillations Orchestrate Multisensory Processing. *The Neuroscientist*, 24(6), 609–626. <http://doi.org/10.1177/1073858418755352>

Miller, J. (1982). Divided attention: evidence for coactivation with redundant signals. *Cognitive Psychology*, 14(2), 247–279.

Murray, M. M., Brunet, D., & Michel, C. M. (2008). Topographic ERP Analyses: A Step-by-Step Tutorial Review. *Brain Topography*, 20(4), 249–264. <http://doi.org/10.1007/s10548-008-0054-5>

Reviewed by Julian Keil

Reviewer #2 (Remarks to the Author):

In this paper De Sousa and colleagues seek to address a major question regarding whether perceptual improvement during multisensory decision is best explained by so-called 'early' sensory processing or 'late' post-sensory change in decision making dynamics. To that aim, the authors employed a visual object categorization task while participants were exposed to either visual-only stimuli or to audio-visual stimuli. They analyzed concurrently recorded EEG data using discrimination analysis to distinguish object categories. Based on this method, the authors selected two EEG components peaking at different latencies: an Early and a Late component. Comparing the two conditions, they found that complimentary auditory information in audio-visual context amplified the Late but not the Early component. Last, they used neurally-informed drift diffusion model and showed that performance gain in multisensory context arose from enhanced single-trial representation of the Late component.

This study is timely and its purpose relevant for the community as it bridges two major fields in neuroscience that are multisensory integration and decision making. The methods used for the analysis are elaborated and appropriate. Nonetheless, several major concerns prevent to endorse the conclusion of the study. Which includes the interrogation whether genuine multisensory integration process is at play during the task. I very much hope that the comments outlined below will be helpful to alleviate some of the issues.

1) The analysis of behavioral data reveals that in this study the multisensory gain comes with slower response times. This result is intriguing as it is not concordant with typical multisensory effects/benefits where faster response times parallel better accuracy. Such unexpected finding raises some interrogations that are not rigorously considered in the manuscript.

Here, an extreme hypothesis can be envisaged: the multisensory context used in this experiment did not lead to multisensory integration per se. For instance because the auditory stimulus is too short – 50ms minus the 10ms on/off ramp – to carry the semantic information that must be integrated with visual information. Under this premise, we could conjecture that participants based their decision either on visual information only or on auditory information only (while the stimuli were short, this later case was made possible because participants learnt during a training session to associate the auditory snippets with the two categories). That is, performance in the multisensory condition increased simply because the complementary auditory information was used whenever visual information did not permit to categorize the stimulus. It followed that when participant based their decision on auditory information response time increased. This increase being caused by modality switch cost in resource deployment and/or by further auditory processing. One way to explore this possibility would be to compare normalized RT distributions (e.g. z-scored by participant) between conditions to assess if the multisensory context leads to a bimodal RT distribution. Another common practice would be to complement the analyses with EEG data recorded for auditory-only trials.

We thank the reviewer for this very thoughtful comment. Indeed, the nature of our experimental design (i.e. learning to associate short sounds with specific visual categories) may have encouraged participants to adopt a strategy in which – on some trials – they only used the complementary auditory information when the visual evidence alone did not allow them to categorize the stimulus (rather than a consistent integration of both pieces of evidence). As the reviewer pointed out, if this was indeed correct, it follows that on this

subset of trials RTs would increase, leading to a bimodality in the RT distribution, which could have been concealed by group differences in the mean and variance of individual RTs.

To rule this out we adopted the reviewer's suggestion and first standardised each participant's RTs (by z-scoring) and then tested the resulting distributions for bimodality formally. Specifically, to test whether the multisensory RT distributions are bimodal, we attempted to fit them with a mixture of exponentially modified Gaussian (expGaussian) distributions, to account for the long tails common in RT data (using maximum likelihood estimation of the pdf parameters). We did that separately for face and car trials and found that one expGaussian fit the data well (adjusted $R^2 = .94$ on average) and that two exponential Gaussians did not fit the data better (adjusted $R^2 = .92$ on average). We also used the Bayesian Information Criterion to compare directly the two fits and found a lower (better) BIC for the model with one expGaussian (BIC1 = -912 vs BIC2 = -688). This result indicates that there was no bimodality present in the RT distributions. The visual inspection of the RT distribution (see below and Fig. 2f) clearly confirms this finding. We now report this result in the manuscript on page 7f. and add the figure below as a new panel f to Fig. 2.

Based on their modeling approach, the authors postulated that longer RTs in the multisensory context reflected additional encoding time needed when auditory processing was at play. However this does not seem to be supported by the electrophysiological results: the discrimination analysis depicted in Figure 3 indicates that in the multisensory condition the Early component peaks earlier. It is only after 300 ms that we can see a sort of slow down (or an acceleration in the visual-alone condition), leading to a delayed Late component in the multisensory condition as to compare to the visual condition. This change in the dynamic is probably at the origin of the slower RTs in the multisensory context. It would be important to further analyze the variation in the processing dynamic to evaluate acceleration/slow down between conditions.

On the recommendation of all reviewers we have now adopted a much more rigorous approach in defining the Early and Late components (rather than basing the selection on a priori assumptions of “older” definitions about the timing and number of components [e.g. Philiastides et al., 2006; Philiastides & Sajda, 2006a; Philiastides & Sajda, 2006b; Philiastides & Sajda, 2007; Ratcliff et al., 2009, Blank et al., 2013, Diaz et al., 2017, etc.]). Specifically, to identify the number of relevant components in the entire range of significant classification as seen in Fig. 3a (180 – 600ms post-stimulus), we employed a temporal clustering approach based on differences in scalp map distributions, which are typically suggestive of changes in the underlying cortical sources/processes (see Materials and Methods for more details). More specifically, we used a k-means clustering algorithm and optimized k (i.e., the number of different time windows with similar scalp topographies). This procedure revealed the presence of two temporally distinct scalp representations with a transition point at 380ms post-stimulus, for both V and AV conditions. These spatial representations are consistent with our previously reported Early and Late components, with centrofrontal and bilateral occipitotemporal activations for the Early and a prominent centroparietal activation cluster for the Late component (Fig. 3a, top). These results remained unchanged regardless of the choice of the evaluation criterion and the distance metric used for the k-means clustering (see Methods on page 33f.).

We then extracted participant-specific component latencies – for each condition separately – by identifying the time points leading to peak A_z (classification) difference within each of the two windows identified by the clustering procedure. Importantly, this entire procedure allowed us to formally evaluate component latencies including all peaks across the entire time course. This analysis revealed no systematic latency differences across V and AV conditions, despite seemingly minor temporal shifts in the group-level average A_z plots, which are likely driven by inter-individual differences in latency and classification accuracy. Importantly with regards to this comment, the peak latencies of the Early component did not differ between V and AV conditions ($M_V = 293\text{ms}$, $SD_V = 53.84\text{ms}$ and $M_{AV} = 293\text{ms}$, $SD_{AV} = 57.52\text{ms}$; paired t-test: $t(39) = 0$, $p = 1$). The peak latencies of the Late component showed no difference in peak timing either ($M_V = 500.25\text{ms}$, $SD_V = 40.92\text{ms}$ and $M_{AV} = 508.25\text{ms}$, $SD_{AV} = 40.76\text{ms}$; paired t-test: $t(39) = -1.09$, $p = .281$). This speaks against the reviewer’s conjecture, which probably was based on the visual appearance of the graphs. We conclude that we do not find evidence for a difference in peak latencies for either EEG component.

The nature of the slower RT in the multisensory context could also be justified further by considering evidence from our neurally informed drift diffusion model. Our model suggests that non-decision times in the AV trials are longer compared to the V trials (by 38ms on average), which points to a prolonged initial encoding and processing of the complementary sensory evidence (possibly due to the auditory stream requiring slightly longer processing times). As highlighted above, this longer stimulus encoding duration may derive from the particular construction of the multisensory context in our task, where decisions are based on a broader audiovisual context and a higher-level conceptualisation of the evidence, such as the presence of car-like objects and general sounds signalling a busy street (rather than fully congruent and hence complementary AV evidence).

These considerations are discussed in more detail in the last two paragraphs of the revised Discussion (pg. 24).

To rule out that the higher recognition performance in audiovisual context was not exclusively related to complimentary auditory information, the authors used a robust bend correlation analysis between subject-specific levels of auditory evidence with individual improvements in decision accuracy on audiovisual trials. This approach might be somehow too indirect as subject-specific auditory noise was estimated to support perithreshold performance. It could be more straightforward to directly correlate decision accuracy between both conditions (auditory and audio-visual). Still the conducted analysis revealed that the amount of auditory evidence explained a fraction of the variance in the accuracy across subjects. And, while the R-value from the correlation analysis is provided, the p-value is not indicated.

The reviewer here touches upon an exploratory analysis, which we reported, but which is not directly relevant to our main question. In fact, while our study directly pinpoints EEG components reflecting the multisensory convergence of AV information and driving the single-trial perceptual benefit, our study was not designed to elucidate what precise aspect of the auditory stimulus is driving this multisensory benefit. Using this specific analysis, we show that the overall amount of acoustic signal (above background noise) is not the driving factor.

Nonetheless, as per the reviewer's suggestion, we would like to offer a second analysis addressing this question. Specifically, we computed a correlation between V and AV decision accuracy collapsed across coherence levels (i.e., one value per participant). This additional analysis addresses the question whether participants who performed well in the visual condition also performed well in the audiovisual condition or whether low performers in the visual condition improved more in the audiovisual condition. The dotted line in the figure below illustrates the line of no improvement. Our results demonstrate two things: 1) a strong correlation between performance in the V and AV condition ($r(38) = .64, p < .0001$; see figure below/ Fig 2c), which underlines the successful selection of auditory noise levels around individual perithreshold performance (i.e., we avoided introducing a bias in the individual amount of available auditory evidence) and 2) most data points fall above the main diagonal (line of no improvement), which provides another illustration of the overall benefit of the complementary auditory evidence across participants. We now present these results on page 7 of the manuscript and include the figure below as a new panel c in Fig. 2.

Another argument supporting the possibility that no multisensory integration is at play in this study relates to the absence of inverse effectiveness effect. The principle of inverse effectiveness in multisensory integration states that, as the responsiveness to individual sensory stimuli decreases, the strength of multisensory integration increases. Accordingly, one may assume that multisensory benefit would be linked to the level of visual evidence. This was tested and, unfortunately, no significant interaction was found between modality and the level of visual evidence.

The reviewer uses the principle of inverse effectiveness (IE) to argue against the absence of multisensory integration in the present paradigm. We find this argument difficult to follow for a number of reasons. While IE had been propagated as a cornerstone of neural multisensory integration by Meredith and Stein in their pioneering work, their own data has shown that for many neurons the multisensory response enhancement shifts from the classical supra-additive enhancement to a de facto sub-additive regime (Stanford, Quessy, & Stein, 2005), and for many neurons outside the superior colliculus no such inverse relations have been reported. In fact, many multisensory influences were reported to actually reduce activity (e.g., Kayser, Petkov, and Logothetis, 2009; Fetsch, DeAngelis, and Angelaki, 2013). Furthermore, studies on classical paradigms such as audiovisual speech perception argue against a clear IE, with largest multisensory benefits at intermediate levels of stimulus efficacy (Ross et al., 2007). Together with statistical fallacies on concluding on the presence of IE in experimental data (Holmes, 2009) this seems to have reduced the field's enthusiasm of relying on this criterion to decide on the presence or absence of any sort of multisensory influence, yet proper multisensory integration.

At the same time, we note that the behavioural data obtained here fit with the more general notion of multisensory integration used in many contemporary studies (e.g., Fetsch et al., 2013, Cao et al., 2019), which, as spelled out by Keil and Senkowski (2018): defines multisensory integration as "any process involving multisensory processing, in which the neural response is different from responses following the modality-specific responses". Based on this reasoning, we believe that the concept of inverse effectiveness does not provide an objective and sensitive index of whether the acoustic and visual information have been perceptually combined by participants to make their choice.

2) The main claim of the paper relies on the discriminator performance being found significantly higher in the multisensory condition only at the latest time period, considered by the authors as post-sensory decision related processing. This is summarized at the beginning of the discussion where the authors wrote: "... AV trials were accompanied primarily by enhancements in a Late EEG component indexing decision-related processes. In contrast, an earlier EEG component encoding sensory (visual) evidence remained unaffected by the addition of complementary auditory evidence". These statements are somewhat hasty considering the results presented in the paper.

First, regarding the Early and Late components, the authors postulate that the components reflect two functional steps: encoding sensory and post-sensory visual evidence, respectively. However this is not demonstrated and only assumed based on the look of component topographies. Considering the previous studies conducted by some of the authors in the visual domain, additional analysis can be conducted to prove their point, especially in the audio-visual context. The authors may also want to think through the work from other groups who did investigate the link between sensory encoding and decision

formation (for instance Romo and colleagues in primates, or O'Connell and colleagues in humans). Moreover, the discriminant analysis actually revealed three peaks but only the first and the last one were considered. What was the rationale to leave aside the second and highest peak?

We thank the reviewer for these suggestions. As we have highlighted above we have now implemented a rigorous temporal clustering procedure (based on scalp topographies) to identify the relevant number of EEG components and their latencies across individuals. Our new analysis suggests that there are only two temporally distinct components defined at 180 – 360ms and 400 – 600ms post-stimulus, which we termed Early and Late, respectively. Importantly, these epochs are identical for the V and AV conditions. To assess the distribution of component peak latencies across individuals, we attempted to approximate their distribution with a Gaussian Mixture Model. We found that, for the Early component latencies, the optimal number of Gaussians is two (both for V and AV) with means ~240ms and ~330ms respectively, which coincide with the two peaks of the discriminant analysis in the 180 – 360ms window. Thus, these two peaks are likely due to interindividual differences in the onset of the Early component.

Turning to the possibility of these components reflecting distinct functional steps our arguments are many as summarised below and now explicitly included in the revised manuscript (pg. 9f, 13,16f):

1. The relative timing of these components and their distinct spatial representations clearly point to separate spatiotemporal representations. In particular, the transition point between these components, as estimated from our latest clustering analysis (see above), sits squarely at the intersection of what is commonly considered early sensory vs post-sensory representations (i.e. around the transition of sensory evoked to P300-like responses).
2. Another piece of evidence pointing to the role of our Late component in encoding decision-relevant evidence, is provided by the fact that this component (unlike the Early one) evolves from a purely stimulus-locked component and evolves/persists until the conclusion of the decision (i.e. shortly before a motor response is initiated). We are demonstrating this in our revised analysis explicitly by means of a very strong correlation between scalp representations in our Late stimulus- and response-locked components ($r_{\text{bend}}(38) = .88$ for V and $.86$ for AV; compare scalp topographies for Late_S and Late_R in Fig. 3a/4a) as well as by the comparable contributions in their single-trial amplitudes in our neurally-informed modelling.
3. Perhaps most importantly, our neurally-informed modelling approach offers a convincing and mechanistic differentiation between the two components, much along the lines of previous work (from our group as well as others, as suggested by the reviewer). Specifically, both components contribute to the rate of evidence accumulation (drift rate) in our nHDDM but the contribution of the Late component (not the Early) is modulated by the coherence levels of the visual stimuli. This indicates that the amplitude of the Late component is closely linked to the reliability of the sensory information as has been shown in past work, (Philiastides, Ratcliff, & Sajda, 2006; Philiastides & Sajda, 2007; Ratcliff, Philiastides, & Sajda, 2009) which, unlike early sensory encoding, is associated with the accuracy of perceptual choices.

Second, the lack of statistical significance does not permit to reach a final conclusion about the absence of an effect. It could be the case that the relevant signal is too small in the present study to lead to significant difference, or that the effect is only present in a subset of the participants. In Figure 3 part b, the difference between discriminator performance for visual and audio-visual conditions clearly show two peaks at 200 and 300 ms almost reaching statistical significance. This is further confirmed in Figure 3 part c, where we can see that the two peaks at 200 and 300 ms are shared by at least 60% of the participants. These two peaks are also visible in the response-lock analysis presented in Figure 4. Also, it would be interesting to investigate if there are two subsets of participants: the early and the late 'integrators'.

We thank the reviewer for drawing our attention to these observations. Indeed, at first sight it is tempting to question the lack of statistical robustness on some of the observations from these figures. However, as we have demonstrated with our new temporal clustering procedure and extraction of participant- and condition-specific latencies for each component, there is very little evidence suggesting that any of these observations would not withstand any additional scrutiny. Specifically, we found no reliable temporal differences in the onset times of either of our Early and Late components across the V and AV conditions.

Moreover, as we have highlighted in an earlier response above, we have characterised the distribution of the peak component latencies (via a Gaussian Mixture Model) to assess whether they are unimodal or multimodal and found that, for the Early component in particular, the optimal number of Gaussians is two (both for the V and AV conditions) with means ~240ms and ~330ms respectively, which coincide with the two peaks of the discriminant analysis in the 180 – 360ms window. Thus, rather than early and late "integrators", our data merely suggest that some individuals encode the early sensory information faster than others.

These results can be found on page 10 in the revised manuscript.

Minors:

- Several information concerning the auditory stimuli are lacking. For instance, the normalization procedure for the pictures is described in details but not the procedure to normalize the sounds. Also, the amount of sounds per categories is not indicated.

We used 15 different sounds per stimulus (i.e., face/car) category. We now indicate this number in the revised manuscript (line 549 on page 26). The auditory sounds were normalized by their standard deviation. This detail is now included in the Methods section on page 26.

- Can the authors specify the mean amount of trials per condition (plus-minus mean absolute deviation)?

The mean amount of trials per condition across participants that was included in our analysis was $V_{\text{car}} = 178.53$; $V_{\text{face}} = 178.78$; $AV_{\text{car}} = 178.33$; $AV_{\text{face}} = 177.88$. The respective mean absolute deviation was $V_{\text{car}} = 1.74$; $V_{\text{face}} = 1.6$; $AV_{\text{car}} = 2.18$; $AV_{\text{face}} = 2.81$ trials. Note that per

condition 180 trials were originally presented to all participants. These numbers are now reported in the Methods of the revised manuscript (pg. 30).

- In the description of the linear multivariate single-trial discriminant analysis, it is not mentioned if only correct trials were used in the procedure.

All trials were included in the face-vs-car single-trial discriminant analysis, consistent with all of our prior work using this design/method. This is critical in order to obtain the full range of *single-trial* EEG activity (discriminator output values; Y_s) and quantify how each trial individually was “experienced” by the participant (i.e., how the stimulus was encoded). Full trial information is also required for running the neurally-informed HDDM in order to fit both correct and incorrect trials. We now specify this in the Methods section in line 733 on page 32.

- Spatial correlation was computed between the components isolated in the visual-only condition and the audiovisual condition, but this is not detailed in the material and method.

Thank you for this suggestion. We assume the reviewer meant the scalp projections (i.e., the forward models) resulting from our discriminant analyses. We therefore attempted to resolve possible ambiguities by adding a sentence to further clarify the computation of these scalp projections for both discriminant analyses (line 773f, pg. 33).

- In Figure 3 and 4, it would be pertinent to extend the x-axis to cover the entire period of interest. That is to include in both figures (and to indicate graphically) the onset of the stimulus and the mean response time plus some baseline on each side to ensure that the discrimination analysis goes back to chance level. Also the color scale for the topographies is missing.

To address this point, we extended the x-axis of figure 3 to 650ms post-stimulus (corresponding to the median RT) in order to avoid omitting information that could be of potential interest. Our reasoning for not extending the figures' x-axes beyond this point, in the first place was that showing discriminator performance beyond (around) mean RT in the stimulus-locked analysis can be heavily misleading, since (pre)motor activity and the overall variability in RTs (e.g. across the stimulus categories we are discriminating) could become a major confound. Consequently, artifactual discriminator performance might arise due to misalignment of motor preparatory activity (for example if RTs in some individuals happen to be slightly slower/faster in one of the stimulus categories). For this reason, A_z plots are not necessarily guaranteed to return back to “baseline” and any attempt to interpret them would be problematic - though, we note, that in our case the A_z traces have virtually returned back to where the “bootstrap” chance level performance would be, as expected.

In short, discriminator results beyond the median RT in the stimulus-locked analysis could be uninterpretable and it is for exactly this reason that we perform a response-locked analysis to correct for this misalignment. Similarly, however, discriminator results from a response-locked analysis that extend too far back in time (i.e. closer to stimulus onset) would also suffer from the same confound (i.e. due to misalignment of early visually/auditorily evoked responses due to potential differences in RT across conditions). For these reasons, we

present results in the intervals in which we can safely trust the output of the discrimination and offer meaningful interpretations (pg. 11 and 13).

- Figure 5 part a, the graphical representation of the HDDM parameters is confusing. It depicts the estimates of the variance (sigma) for tau and alpha, but this is not mentioned in the text. Moreover, mu and sigma for tau point toward alpha, while mu and sigma for alpha point toward tau.

We thank the reviewer for spotting this error, we now corrected the subscripts on figure 5a. We also revised our Methods section to clarify the use of population means and variances (pg. 37f.).

- Figure 5 part b, RT distributions are substantially different between Car-choice and Face-choice. Is it related to difference in performance between the two categories? Can the authors provide performance and mean RTs for the two categories (+/- m.a.d.)?

Please find the requested information in the following. Car choices are less accurate than face choices: accuracy = 0.72 ± 0.05 vs 0.83 ± 0.07 on average for V trials and 0.77 ± 0.04 vs 0.86 ± 0.06 on average for AV trials. Also, car choices are slightly slower than face choices: RT = 0.632 ± 0.084 vs 0.628 ± 0.068 seconds on average for V trials and 0.672 ± 0.095 vs 0.656 ± 0.066 seconds on average for AV trials.

We now report these results and explain how they are reflected in the obtained HDDM parameters (i.e., starting point and drift rate biases) on page 18.

- At the end of the neurally-informed modelling analysis, the authors underlined the fact that “the average difference in RTs (37ms) is very similar as the average non-decision difference between the two conditions (38ms)”. Yet, did the authors try to correlate the two variables across participants?

Thank you for this comment. As Reviewer 3 pointed out below, as a consequence of using a hierarchical model, the participant-level estimates of the parameters are not independent samples because they are drawn from the same population-level probability distribution. Hence, it would not be correct to treat the non-decision time estimates of each participant as independent samples and correlate them with the participant RTs. For this reason, we have removed a similar correlation we performed between drift rate differences and accuracy differences (previous Fig. 5d).

- In the report of the HDDM constrained only by the behavioral data, the drift rate of evidence accumulation was not provided. Can the authors add this information?

Please note here that our HDDM models choice rather than accuracy, with 1 indicating face choices and -1 indicating car choices. Thus, if there was no bias at all, the average drift rate would be zero. As participants made more car responses than face responses, the average drift rate is shifted towards a negative value (favouring car choices). This result is consistent with the reviewer’s comment on the difference between face and car RT distributions, namely the higher number of car choices is explained by a higher drift rate for car choices. We now included this information on page 18 of the revised manuscript.

Reviewer #3 (Remarks to the Author):

In their paper “Auditory information enhances post-sensory, but not early sensory, visual evidence during multisensory decision making”, De Sousa and colleagues perform an EEG study of multisensory perceptual choice and combine this with cognitive modeling to investigate how and when the benefit of multisensory information impacts on the choice process. Using an EEG discriminability analysis established by the senior author, they identify an early and a late EEG component that allow to separate visual evidence for the two choice options (face vs. car). Only the late component distinguishes between the multisensory and the purely visual conditions in the task, both stimulus- and response-locked. The authors go on and combine the EEG data with hierarchical modeling of the drift diffusion model (DDM) and find that only the weight of the late EEG component on the drift rate is stronger in the multisensory condition. Furthermore, they show that fitting the DDM without neural data

could lead to different (and possibly wrong) conclusions (i.e., a higher boundary separation in the multisensory condition).

Overall, the paper is written in a very clear, succinct and precise style, and the study addresses a timely research question (i.e., the computational and neural principles of multisensory perceptual decision making). The related literature is introduced and discussed sufficiently, the methods and results are presented clearly, and the potential impact of the current study is put into context adequately. Given that there have already been some publications on modeling multisensory integration with the DDM (e.g., Drugowitsch et al., 2014, eLife) and on EEG patterns of multisensory integration, the novelty of the current work lies in the sophisticated combination of EEG with cognitive modeling. I agree with the authors that this provides potentially interesting and important new conclusions, in particular because the modeling results seem to differ substantially depending on whether the neural data is included or not. However, I have several remarks on the manuscript, most of them

of methodological nature, that need to be addressed to ensure that the modeling (but also the EEG) conclusions are indeed justified.

Major points:

1. Cognitive modeling: When estimating the DDM, the authors seem to ignore a potential starting point bias. In the methods, they state that the task would not induce such a bias and refer to previous work. However, Figure 5b seems to indicate that car choices were more frequent than face choices, which could speak for such a starting point bias effect. I might be wrong here (descriptive statistics about the choice frequencies would be helpful), but if not, I think this possibility should be accounted for by treating the starting point as a free parameter in the DDM.

We thank the reviewer for this insightful suggestion. We now allow the starting point to vary to test if it accounts for the higher proportion of car choices. We should note here that choice proportions can be modelled in the DDM in two ways: a) by moving the starting point closer to the most frequent response, or b) by allowing for different drift rates for each choice, i.e. shifting the drift rate towards the most frequent response (drift bias as in Ratcliff, 2008). Here we found

- a) an average starting point of 0.55 towards face choices (expressed as a proportion of the boundary separation, i.e. taking values between 0 and 1) and
- b) an average drift rate that is shifted towards a negative value favouring car choices (with positive values indicating face choices and negative values indicating car choices). Please note here that our HDDM models choice rather than accuracy, with 1 indicating face choices and -1 indicating car choices.

Hence, we deduced that the higher number of car responses was explained by a higher drift rate in car trials compared to face trials. This can be understood better if we look at the RT distributions for face and car choices. Car RTs appear to have longer tails but similar modes to the face RTs, which is consistent with a larger drift rate for car trials and not a bias in the starting point, which would shift the whole RT distribution. On the contrary, we found a small starting point bias towards face choices. This can be explained by the lower accuracy in car choices compared to face choices (0.74 vs 0.84% on average). In other words, more errors occur when the starting point is further from the boundary where a response would be correct. Thus, this combination of RTs and choice accuracies resulted in higher drift rate for car choices and a starting point bias towards face choices. Overall, these two biases together with the use of the neural information in the model enabled us to obtain a good fit of the behavioural data by the model parameters. Crucially, as these two biases are consistent across V and AV conditions, they do not affect the interpretation of our findings, attributing the multisensory benefit in behaviour to late post-sensory mechanisms. We present and discuss these findings on page 18 of the revised manuscript.

Second, I was surprised that the different visual conditions were fit independently from each other rather than assuming a “linking” function (e.g., linear, power, logistic) that connects phase coherence with the drift rate (like in Philiastides et al., 2014, J Neurosci). First, this should just be more efficient. Second, the current approach apparently yields a beta weight of the EEG data on the drift rate for each coherence level (see methods). How do you integrate those then within a participant? Taking the mean across coherence levels? Furthermore, in the methods it says that statistical inference on the beta weights were made by looking at the posterior density (i.e., in a Bayesian way), but frequentist statistics are reported in the results (page 14). In my view, the best approach would be to assume that single-subject beta weights are drawn from (normally distributed) group-level distributions for means and SDs, and then to perform the statistical tests on those group-level means (see Boehm et al., 2018, Behav Res Method).

This is a very useful suggestion that enabled us to improve our model and the way we reported our results. First, as the reviewer suggested, we now modelled drift rate as a linear function of phase coherence. We used different models testing the effect of coherence on drift rate and found that the following model provided the lowest DIC:

$$DR(i) = \gamma_0(i) + \gamma_1(i) * Y_{Early}(i) + \gamma_2(i) * Y_{Late}(i) * coh(i), i=1, \dots, K \text{ trials}$$

This suggests a modulation of the Late component amplitudes with coherence to scale the drift rate in single trials, consistent with the role of this component in indexing the quality of the evidence entering the decision process as in our previous work.

We also followed the reviewer's suggestion to estimate group-level distributions for the beta weights and assessed significance by comparing the group-level distributions in V and AV, which were assumed to be Gaussian with means defined by the group-level estimates for $\mu_{\delta}(V)$ and $\mu_{\delta}(AV)$ and variance $\sigma^2 = (\sigma^2(V) + \sigma^2(AV))/N$, where N is the number of participants.

We found that $\gamma_2(AV)$ was significantly higher than $\gamma_2(V)$ and no difference between $\gamma_1(AV)$ and $\gamma_1(V)$. In sum, our results suggest that both EEG components are predictors of the rate at which evidence is accumulated. Importantly, the Late component represents the amount of evidence entering the decision evidence, which affects the accuracy of choices. The differential activation of the Late component between V and AV leads to higher evidence integration in AV, ultimately leading to better behavioural performance in AV.

All the above have been amended in both the Methods and Results sections (pg. 16ff. and 37ff.).

Third, a more precise specification of the priors is required.

We chose non-informative (uniform or Gaussian) priors for the group-level means and standard deviations (SDs). All participant-level parameters are drawn from the group-level Gaussian distributions with the above means and SDs. We now specified the prior for each parameter in the Methods section on page 37.

2. EEG analysis: I am wondering whether the fact that the audiovisual and the visual conditions induced differently long non-decision times, and in particular differently long sensory encoding times, could pose a conceptual problem for the EEG analysis. The early and late components are compared to each other in a stimulus-locked way at every point in time, but if the evidence accumulation process starts later in the audiovisual condition, then the question is whether comparing the same time points across conditions actually makes sense. Obviously, this concern does not affect the response-locked analyses.

In general, the reviewer is correct in saying that latency differences in the onset of certain neural processes may lead to issues in the context of a stimulus-locked analysis.

On the recommendation of all reviewers we have now adopted a much more rigorous approach in defining the Early and Late components (rather than basing the selection on a priori assumptions of "older" definitions about the timing and number of components [e.g. Philiastides et al., 2006; Philiastides & Sajda, 2006a; Philiastides & Sajda, 2006b; Philiastides & Sajda, 2007; Ratcliff et al., 2009, Blank et al., 2013, Diaz et al., 2017, etc.]). Specifically, to identify the number of relevant components in the entire range of significant classification as seen in Fig. 3a (180 – 600ms post-stimulus), we employed a temporal clustering approach based on differences in scalp map distributions, which are typically suggestive of changes in the underlying cortical sources/processes (see Materials and Methods for more details). More specifically, we used a k-means clustering algorithm and optimized k (i.e., the number of different time windows with similar scalp topographies). This procedure revealed the presence of two temporally distinct scalp representations with a

transition point at 380ms post-stimulus, for both V and AV conditions. These spatial representations are consistent with our previously reported Early and Late components, with centrofrontal and bilateral occipitotemporal activations for the Early and a prominent centroparietal activation cluster for the Late component (Fig. 3a, top). These results remained unchanged regardless of the choice of the evaluation criterion and the distance metric used for the k-means clustering (see Methods on page 33f.).

We then extracted participant-specific component latencies – for each condition separately – by identifying the time points leading to peak A_z (classification) difference within each of the two windows identified by the clustering procedure. Importantly, this entire procedure allowed us to formally evaluate component latencies including all peaks across the entire time course. This analysis revealed no systematic latency differences across V and AV conditions, despite seemingly minor temporal shifts in the group-level average A_z plots, which are likely driven by interindividual differences in latency and classification accuracy.

Specifically, the identical transition point of the two components around 380ms post-stimulus for visual and audiovisual forward models as well as virtually no difference for the Early ($M_V = 293\text{ms}$, $SD_V = 53.84\text{ms}$; $M_{AV} = 293\text{ms}$, $SD_{AV} = 57.52\text{ms}$) and Late ($M_V = 500.25\text{ms}$, $SD_V = 40.92\text{ms}$; $M_{AV} = 508.25\text{ms}$, $SD_{AV} = 40.76$) components' peak times argue against latency shifts posing a problem for the employed stimulus-locked sample-based analysis. In addition, *we performed a robust bend correlation analysis (Pernet, Wilcox, & Rousselet, 2013) to test the topographical consistency between the Late stimulus-locked and response-locked components. We computed the average scalp map (i.e., forward models) across participants at the point of peak discrimination for the two components (500ms post-stimulus and 100ms pre-stimulus, respectively) and assessed their similarity by computing their correlation (pg. 35f). We found a very high similarity between stimulus- and response-locked forward models of the Late component ($r_{\text{bend}(38)} = .88$ for V and $.86$ for AV; compare scalp topographies for Late_S and Late_R in Fig. 3a/4a), which further supports the notion that they share common underlying generators, consistent with the idea that the Late component persists until the time of the eventual decision.*

These results are presented on pages 10 and 13 of the revised manuscript.

I am further wondering whether the statistical requirement of having 3 consecutive time points at which the two conditions must differ at $p < .05$ is really a proper way of correcting for multiple comparisons. Is this just a rule-of-thumb (like $p < .001$ and 10 contiguous voxels as seen in many fMRI studies) or is it based on a more principled way of correcting for multiple comparisons (like an FWE- or FDR-correction in fMRI)? Perhaps the authors can elaborate on this.

We would like to thank the reviewer for this useful clarification request. We have now introduced a formal procedure to derive a cluster-based correction directly from our data (in the spirit of the Maris & Oostenveld paper). Specifically, we computed multiple null distributions of our A_z data by first applying a permutation procedure (i.e., shuffling temporal samples without replacement) to all temporal samples on the individual participant level, whereby we abolished the relationship between temporal samples. Importantly, by only shuffling the temporal order of the samples but not the difference scores between our

modality conditions for each sample, the relative difference between unisensory (V) and multisensory (AV) A_z values remained unchanged for each sample and participant. This permuted data served as input to a subsequent sample-based bootstrapping procedure, which was identical to the bootstrapping procedure applied to obtain the median difference scores in our original analysis. Using this procedure, we computed the maximum number of adjacent significant samples of the largest cluster for each iteration and stored this value to build a null distribution of maximum cluster sizes. We repeated these steps 1000 times and selected the value corresponding to the 95th percentile of this distribution as minimum cluster size threshold for comparisons with clusters of significant samples in the original data. This data-driven, cluster-based correction for multiple comparisons yielded an average minimum cluster size of three consecutive samples for stimulus- and response-locked data quantifying the difference between V and AV A_z traces. More details of this procedure are now given in the Methods section on page 34f. (line 773ff.).

3. Although they cite the study by Drugowitsch et al., 2014, eLife, I think this study and its comparison to the current results deserve a more thorough discussion. The behavioral findings are very different: in the Drugowitsch paper, participants make faster and sometimes even worse decisions in the multisensory case; in the current case, they are slower and more accurate. To me, this appears as if participants used different strategic approaches in the two studies. Why could that be the case? And is this interpretation in line with the current modeling results (which seem to speak against a threshold-adaptation account)?

We thank the reviewer for this suggestion, we now discuss our results in relation to the proposed paper as follows (pg. 22):

Similarly, reaction time differences could arise due to the choice of sensory modalities and/or inter-individual choice strategies employed by participants. For example, a recent study (Drugowitsch, Deangelis, Klier, Angelaki, & Pouget, 2014) using time-varying multisensory information (visual and vestibular) reported faster but slightly less accurate choices compared to unisensory decisions. The authors modelled these results with a novel variant of the DDM model that incorporates the effects of time-varying information and sensory cue reliability and reported consistent drift rate improvements in the multisensory condition across participants. In other words, despite differences in the behavioural outcomes, this study and ours, both suggest that multisensory information leads to faster integration of sensory evidence.

Minor points:

4. Statistical analysis of the effects shown in Fig. 5c: I think it is more appropriate to test for effects of sensory conditions, early/late components and their interaction using an ANOVA rather than multiple paired t-tests.

We agree with the reviewer on this point. However, following the reviewer's suggestion above, we now altered our model by scaling the Late component amplitudes by the single-trial coherence levels. It follows that the two beta's are not directly comparable in this new

model. Therefore, we used separate “hierarchical” t-tests at the population level (as suggested by the reviewer) to test the effects of modality conditions (V vs AV).

5. How was the sample size determined?

Sample size standards in the field when EEG measurements are employed normally range from 20 to 60 participants. An a priori power analysis for a fixed linear multiple regression model with 2 predictors, a medium effect size of 0.5 (typical for research in psychology), an alpha of .05, and a power of .95 resulted in a required sample size of 35 participants. Hence, we surpassed the determined sample size to have more power than was determined as a minimum. We present this analysis in the Methods section on page 25 of the revised manuscript.

6. Typos: line 177 (“as” rather than “a”); line 315 (“similar to” rather than “similar as”); line 327 (“a” rather than “as”); line 359 (delete the comma after “specifically”); line 721 (“face.vs.car”).

Thank you for spotting these typos, which have been corrected where appropriate in the revised version of the manuscript.

Signed,
Sebastian Gluth

Reviewers' Comments:

Reviewer #1:

Remarks to the Author:

The authors have taken great care to address all issues. Accordingly, the already good manuscript was further improved. Yet, some minor issues still remain. In general, I advise highlighting all changes to a manuscript to ease the review process and facilitate spotting editing errors.

Line 537: Could you include the link to the image database here?

Line 593: Please double check for editing errors here.

Line 872: Do these values refer to the response-locked component? If so, please state more clearly. However, the interval differs from the interval described in line 254, so I may have missed something.

Line 116: Here, I'd state the comparison, e.g. "more accurately than on V trials".

Signed: Julian Keil

Reviewer #2:

Remarks to the Author:

Even though the authors tried to address the issues raised in the previous round of review, the main concerns have not been answered satisfactorily. First, the authors unnoticed the fact that the behavioral effects are not explained by the conducted analyses. Second, the experimental design does not allow to alleviate some potential cofounds and some controls should be performed. Third, the introduction tends to disregard seminal works produced by other researchers. Last, some findings are not taken into account in the discussion. I develop these different points down below and I hope it will be constructive.

1) Considering the claim of the study, the key issue relates to the lack of explanation of the multisensory behavioral effects (i.e. better accuracy and slower response time in the multisensory condition). First, accuracy was not taken into account in the investigation, neither in the linear discriminant analysis nor in the nHDDM (in both approaches, all trials were included without distinction between correct and incorrect responses). Second, the slower AV RT is far to be explained. In the decoding procedure two components were identified, an early and a late component, showing no significant difference in their latency peak between the visual-alone and the AV condition. In the nHDDM, which aimed at modeling RT distribution, drift rate was found to be higher in the AV condition; this result typically translates into faster RT, which is at odds with the reported slower AV RT.

The RT slowdown on AV trials is interpreted as an additional encoding time required for the auditory stimulus. But, while the authors link the early component to sensory evidence encoding, the latency peak of that component does not differ between AV and V condition. Also, the nHDDM shows significantly longer ' non-decision time ' in AV trials, which is again interpreted as reflecting extra time to process auditory stimulus. However this ' non-decision time ' parameter includes both stimulus encoding and response production. Based on the latency analysis, a more parsimonious explanation leans toward a difference in the motor response preparation; rather than a difference in the brain dynamic prior to the making of the decision, as assumed in the manuscript. These inconsistencies may motivate the authors to revise their statements.

2) In link with my comments from the previous review, it is doubtful that genuine multisensory integration is at play in this experiment. The observed difference between the visual-alone condition and the audiovisual condition can arise from, and be explained by, unspecific effects, such as memory recollection, resource deployment and/or difference in attentional set. Unfortunately, control conditions are missing and would be helpful to verify authors' conjectures.

The authors utilized an experimental design that is almost identical to the one they have been using to investigate visual perceptual decision making. This choice is rather questionable. In the auditory domain information is carried out by time; while in the visual domain all information is accessible at once, allowing fast picture presentation. In this experiment, the use of brief sounds (50 ms minus 2 x 10ms on/off ramp) necessitated a training session with visual feed-back to permit a correct categorization of the sounds. As a consequence, the learned associations introduced a recollection dimension to the main experiment and did not favor fast, and natural, auditory categorization. As opposed to the visual-alone condition, in the audiovisual condition the recall/retrieval of the learnt association involves additional resource(s), slowing down decision making process and therefore biasing the difference between unisensory and multisensory conditions toward later latencies. Per construct the task and the linear discriminant analysis are based on visual categorization, and hence the non-attended auditory information is not prioritized. Here, some controls are needed, for instance by asking to the participants to equally pay attention to both modalities and/or to evaluate the decoding in an auditory-alone condition and/or to use longer stimuli (i.e. more ecologic).

In the training session, the participants had to learn to associate the very short auditory stimuli with one of the two categories. Actually, to perform the training session, learning to match sounds to only one category is sufficient. Like learning to recognize the human speech sounds only. Even normalized and embedded in some noise, human speech sounds are more homogeneous in their spectral contents than the car/street related sounds (that were squeaking tires or slammed door sounds). This disparity between categories may provide some explanation to the difference observed in the accuracy as well as in the responses time distribution, that were paralleled by the results of the nHDDM (i.e. starting point bias) . In link with this point, it would be important to have access to the associated statistical comparisons (i.e. difference in accuracy and RT between categories as well as conditions).

3) Some of the findings ask for further examination to assess how they may have influenced the main results and/or to clarify their meaning.

For instance, a latency analysis performed on the Early component demonstrated the existence of two groups in the participants. Since this is an unexpected result, it calls for additional analysis and should be incorporated as a variable of interest in the analyses performed. As an example, how many participants belong to each group? Does this two groups correspond to the participants for which the Late component amplitude is higher/lower in the AV condition as to compared to the V condition?

Another illustration relates to the drift rate difference between the Early and the Late component (Figure 5). It is interesting to see that the drift rate is larger than zero (Result section line 342). But even more interesting to note that for the Early component the parameter is between 0-1, and therefore reduces the impact of the Early component on the drift rates; which is not the case for the late component with a parameter above 1 in average.

In link with this point, since the authors linked the Early component to sensory encoding (see lines 317, 339-344), it would make more sense to inform the non-decisional time (rather than the drift rate) with the amplitude of the early component.

Last, the authors began to envisage some of the issues I raised in the previous review, but their rapid

analysis is only superficial (e.g. they evaluated if the low performers in the visual condition improved more in the audiovisual condition). To pursue in that direction, it would be pertinent to take into account coherence levels to assess if the participants tend to use more the auditory information when the image is noisier.

4) In the introduction, the authors undermine the current knowledge of the field. For instance in the fourth paragraph (lines 59 to 70), all the cited neuroimaging studies were published before 2013; even though some recent publications combined modeling and MEEG data (for instance, Rohe et al 2019 or Cao et al. 2019, cited later in the manuscript and publish in Nat.Com.). Also, it would be informative for the community to have some links with more classical approaches. Here, one may think about the ERP community where some components have been linked to specific steps of decision making process (see for example the seminal work of O'Connell and colleagues).

Minors comments:

Abstract

Line 17: "'early' sensory processing benefits" can the authors clarify what they mean by early benefits?

Line 25: "multisensory behavioural improvements" The plural form is not correct here as only accuracy was improved, not RT.

Line 27: "high-level conceptualization" this is vague term, it should be clarified.

Materials and Methods

Lines 594-5: A sentence has been truncated.

Lines 707-709: This part of the sentence is confusing 'visual evidence in each modality'. The word 'condition' would suit better than the word 'modality'.

line 872: Does the negative sign means that the range for the late component is computed relative to the response?

line 875: Can the authors rationalized why the two components amplitude are not processed the same way since the later one is multiplied by C.

Results:

Line 160: To illustrate a correlation it is recommended to use square axis with identical scaling.

Line 196: For the early component, the mean peak times for the V and AV conditions appear to be exactly the same, down to the last decimal! This should be underline to ensure the reader that it is not a mistake or an oversight.

Line 331: This result is not surprising since the authors already show the link between stimulus phase coherence and the late component. This original finding should be reminded. As such this result only

confirms the dissociation between the Early and Late component.

Line 363: These results should be represented in a dedicated figure to show to the reader how parameters differ between categories per conditions (please include the variance, like using the mean absolute deviation).

Line 365: The corresponding values should be provided.

Line 369: The bias in the starting point is rather the origin, not the outcome, of the accuracy difference.

Line 374: This statement is hurried; values and statistical tests must be provided.

Discussion

Lines 455-462: This paragraph lacks of clarity, can the authors make their point clearer?

Reviewer #3:

Remarks to the Author:

The authors have done an impressive job in addressing my concerns about the previous version of the manuscript.

Signed,
Sebastian Gluth

We would like to thank the reviewers and editors for their time and feedback. We have compiled our answers to all points raised by the reviewers below (in blue) as well as tracked our changes in the revised manuscript.

REVIEWER COMMENTS

Reviewer #1 (Remarks to the Author):

The authors have taken great care to address all issues. Accordingly, the already good manuscript was further improved. Yet, some minor issues still remain. In general, I advise highlighting all changes to a manuscript to ease the review process and facilitate spotting editing errors.

Line 537: Could you include the link to the image database here?

We have included the link to the image database (<https://www.mphiliastides.org/files/download/228>) in line 599 of the revised manuscript.

Line 593: Please double check for editing errors here.

We have revised the sentence in question to provide additional clarification about the feedback participants received during training. Now the sentence reads as follows:

“During the training session, participants also received visual feedback following each response (on all three tasks). Feedback was presented centrally for each of the three possible outcomes; ‘Correct’ written in green, ‘Incorrect’ written in red and ‘Too slow’ written in blue (when participants exceeded the response deadline).” (pg. 30, lines 654-657)

Line 872: Do these values refer to the response-locked component? If so, please state more clearly. However, the interval differs from the interval described in line 254, so I may have missed something.

Since the last revision we have used the Late stimulus-locked component amplitudes for the modelling as stated in lines 332-336 of the results section (for consistency and direct comparisons with the Early stimulus-locked component). Line 872 was carried over from the original manuscript (referring to the original definition of our Late response-locked component) and should have been amended in the last revision (many thanks for spotting this!). We have now corrected the relevant sentence as follows:

“[...] the single-trial discriminator amplitudes of participant-specific stimulus-locked Early EEG components (individual peak A_2 across V and AV in the time range 180 – 360ms post-stimulus) and stimulus-locked Late EEG components (individual peak A_2 difference between AV and V in the time range established in Fig. 3b; 490 – 540ms [expanded further by 30ms on either side to account for the resulting A_2 values being obtained with 60ms training windows centered on these times]).” (pg. 42, lines 945-949)

Line 116: Here, I'd state the comparison, e.g. "more accurately than on V trials".

Thank you for this suggestion. We have amended the sentence as requested:

"We found that participants performed more accurately on AV than on V trials." (pg. 7, lines 113-114)

Signed: Julian Keil

Reviewer #2 (Remarks to the Author):

Even though the authors tried to address the issues raised in the previous round of review, the main concerns have not been answered satisfactorily. First, the authors unnoticed the fact that the behavioral effects are not explained by the conducted analyses. Second, the experimental design does not allow to alleviate some potential confounds and some controls should be performed. Third, the introduction tends to disregard seminal works produced by other researchers. Last, some findings are not taken into account in the discussion. I develop these different points down below and I hope it will be constructive.

1) Considering the claim of the study, the key issue relates to the lack of explanation of the multisensory behavioral effects (i.e. better accuracy and slower response time in the multisensory condition). First, accuracy was not taken into account in the investigation, neither in the linear discriminant analysis nor in the nHDDM (in both approaches, all trials were included without distinction between correct and incorrect responses).

The reviewer touches on an important question: would our conclusions change if we specifically focused on correct trials only? We performed additional analyses to rule this out, and we can safely say that our conclusions hold regardless of whether we consider all trials (as in the manuscript) or specifically focus on correct trials only.

We separated the individual peak component amplitudes (y) of correct and incorrect trials (Figure 3d and 4d) and compared these between V and AV conditions using Bayesian general linear mixed effects models with participants as a random variable. Contrasting models with and without sensory modality as additional predictor of stimulus- and response-locked data, we found strong evidence for including modality as additional predictor ($BF_{10} = 10.27 \pm 0.9\%$; $BF_{10} = 506.3 \pm 0.73\%$, respectively). This provides (1) strong evidence for higher component amplitudes on AV compared to V trials on correct as well as incorrect trials (alternative hypothesis; see figure below), and shows (2) that component amplitudes were overall higher on correct compared to incorrect trials (in line with the established role of this component in indexing overall decision evidence). The consistency of higher component amplitudes on AV trials (effect of sensory modality) across decision accuracy was confirmed further by our data providing more evidence for a model without an additional modality-by-accuracy interaction term ($BF_{10} = 2.06 \pm 1.3\%$; $BF_{10} = 3.05 \pm 0.86\%$, respectively). This confirms that the reported modality effect (i.e., higher component amplitudes on AV trials) holds regardless of decision accuracy and time point the analysis was locked to (figure a-d below). We have now added these results to the manuscript (lines 241-245 and 272-276, respectively) and figures 3 and 4 (as panels 3e and 4e, respectively).

These results also reinforce that the component amplitudes (y) are a suitable index of the amount of neural evidence. Indeed, in the nHDDM analysis the drift rates (representing the quality of the available decision evidence) for correct trials are on average higher than those of incorrect trials ($\delta_{\text{correct}} = 0.27 \pm 0.09$ vs $\delta_{\text{incorrect}} = 0.14 \pm 0.11$ for face choices and $\delta_{\text{correct}} = -0.65 \pm 0.11$ vs $\delta_{\text{incorrect}} = -0.34 \pm 0.10$ for car choices - with the convention of positive signs for faces and negative signs for cars). This finding provides strong support that accuracy effects are effectively directly captured by our nHDDM (lines 360-364 in the manuscript). In summary, these additional results corroborate the notion that the selected subject-specific peak amplitudes are reflective of the amount of neural evidence available for the decision at play. They confirm that the neural effects across all trials presented in the (revised) manuscript are not driven by decision accuracy.

Second, the slower AV RT is far to be explained. In the decoding procedure two components were identified, an early and a late component, showing no significant difference in their latency peak between the visual-alone and the AV condition. In the nHDDM, which aimed at modeling RT distribution, drift rate was found to be higher in the AV condition; this result typically translates into faster RT, which is at odds with the reported slower AV RT.

The RT slowdown on AV trials is interpreted as an additional encoding time required for the auditory stimulus. But, while the authors link the early component to sensory evidence encoding, the latency peak of that component does not differ between AV and V condition. Also, the nHDDM shows significantly longer ' non-decision time ' in AV trials, which is again interpreted as reflecting extra time to process auditory stimulus. However this ' non-decision time ' parameter includes both stimulus encoding and response production. Based on the latency analysis, a more parsimonious explanation leans toward a difference in the motor response preparation; rather than a difference in the brain dynamic prior to the making of the decision, as assumed in the manuscript. These inconsistencies may motivate the authors to revise their statements.

While this comment suggests a different interpretation of the non-decision time as we presented in the manuscript, it comes without a specific reasoning as to why one could expect differences in motor components between V and AV trials, in particular given that the response type (button press) and motor effectors were identical across both modalities. Hence, we can only speculate as to why the reviewer raises this concern.

Firstly, we assume the reviewer interpreted the lack of latency differences in our Early (sensory) EEG component across V and AV trials as evidence that the observed non-decision time increases during AV trials (which captures both early sensory encoding and motor preparation times) are due purely to motor preparation. This presumption is problematic since our Early (sensory) EEG component represents 'visual' evidence, specifically. In other words, if there was an increased sensory encoding time due to the processing of the added 'auditory' evidence in AV trials, this would still be captured as an increase in non-decision time in the model, even in the absence of a latency shift in the encoding of the 'visual' evidence. In fact, this is the main contribution of our work: during rapid decision-making, the added auditory information influences the encoding of Late (post-sensory) visual evidence but not the encoding of Early (sensory) visual evidence.

In line with this conclusion, the observed difference in non-decision times tracks the difference in RTs closely only when we incorporate neural evidence in our model. Using a version of the DDM model constrained only by behavioural data led to non-decision times not resembling the observed difference in RTs (~19ms instead of ~33ms; lines 425-432). In our view this discrepancy illustrates that the difference in non-decision times arises specifically from the evidence provided by the neurally informed model. In other words, if the non-decision times were merely due to differences in motor preparation as the reviewer speculates, they should not have differed between the neurally vs behaviourally constrained variants of our model.

Instead, as we already discussed in the manuscript (pg. 24, lines 499-504) our findings favour an explanation of the increase in RTs as arising from additional resources required to analyze the sounds. In the present paradigm the auditory information is rapid, context dependent, and complementary (rather than redundant) to the visual information. Therefore, the sounds in our task are treated as supplementary evidence that require deployment of additional processing resources, instead of simply providing confirmation of the evidence received in the other modalities.

Yet, to accommodate the reviewers' concerns we have highlighted the points above by adding this to the discussion of the revised manuscript (lines 506-512).

2) In link with my comments from the previous review, it is doubtful that genuine multisensory integration is at play in this experiment. The observed difference between the visual-alone condition and the audiovisual condition can arise from, and be explained by, unspecific effects, such as memory recollection, resource deployment and/or difference in attentional set. Unfortunately, control conditions are missing and would be helpful to verify authors' conjectures.

We find the allusion to unspecific effects such as memory recollection, resource deployment and/or difference in attentional set as potential confounds very generic, and coming without a clear and specific argument as to why any such effect should play a major role here. We consider it highly unlikely that any of the mentioned factors are at play, in particular as we carefully provided equal training on all visual and auditory stimuli as well as clear instructions to avoid introducing any such confounds. More importantly, our methodology and general discriminant analysis approach is designed specifically for dealing with such unspecific effects. We are elaborating on the reasons below as well as in the revised manuscript (pg. 24, lines 488-496).

Firstly, we would like to reiterate that the discriminant component amplitudes (y) are a weighted reflection of all available neural evidence with respect to the specific decision task (face vs car) we asked participants to perform. These component amplitudes reflect this decision-specific difference in neural signals that resulted from separate V and AV discriminant analyses. Since differences in unspecific effects should have been present on all trials of a given stimulus type and modality they would not have contributed to the estimation of the relevant classification weights. Put simply, in our multivariate analysis framework, these unspecific effects will be subtracted out since they will be present in both face and car trials. This is true for both the V and AV modalities. We view the deployment of multivariate analysis methods as another strength of our work over conventional ERP analysis that is more susceptible to such confounds. A description of this advantage of our multivariate analysis was added to the manuscript's Methods section (pg. 35, lines 784-788).

More broadly, whether attentional demands are comparable across A and AV conditions can be debated. However, theories about a role of attention in multisensory integration propose the rather automatic binding of stimuli across modalities when these are spatio-temporally synchronized (Talsma, Senkowski, Soto-Faraco, & Woldorff, 2010). As well, recent work suggests that the influences of multisensory information and attention operate independently across cortical columns (Gau, Bazin, Trampel, Turner, & Noppeney, 2020). Furthermore, a recent review by Zuanazzi and Noppeney (2020) concluded that attentional resources are largely shared across sensory modalities, hence arguing against a competition between sensory modalities for attentional resources. To ensure no difference in attentional set, all trials were randomized across sensory conditions so that participants were equally likely to encounter (and expect) a V or AV stimulus on any given trial. For all these reasons, we feel confident that the unspecific effects mentioned by the reviewer would not have given rise to the effects reported in the manuscript. To ensure full transparency, we added a brief paragraph on this matter to the discussion of the revised manuscript (pg. 24, lines 488-496)

Relating to the use of the term "multisensory integration" we would like to highlight that it is indeed used somewhat loosely (i.e., in different ways) in the literature. To acknowledge this ambiguity as well as the specific nature of our task, we have removed any reference to the term 'multisensory integration' (as it relates to our own findings) from the manuscript. Instead, we adopted versions of the term

'multisensory combination of additional evidence' consistently throughout the revised manuscript. This wording aims to reflect that the additional information provided by the auditory sensory channel is directly reflected in the larger amount of neural evidence on those trials, as illustrated by the subject-specific peak component amplitudes (y). It is also doubtful that the larger amount of neural evidence we observed would be possible without any sort of combination of neural evidence from both sensory channels.

The authors utilized an experimental design that is almost identical to the one they have been using to investigate visual perceptual decision making. This choice is rather questionable. In the auditory domain information is carried out by time; while in the visual domain all information is accessible at once, allowing fast picture presentation. In this experiment, the use of brief sounds (50 ms minus 2 x 10ms on/off ramp) necessitated a training session with visual feed-back to permit a correct categorization of the sounds. As a consequence, the learned associations introduced a recollection dimension to the main experiment and did not favor fast, and natural, auditory categorization.

As opposed to the visual-alone condition, in the audiovisual condition the recall/retrieval of the learnt association involves additional resource(s), slowing down decision making process and therefore biasing the difference between unisensory and multisensory conditions toward later latencies. Per construct the task and the linear discriminant analysis are based on visual categorization, and hence the non-attended auditory information is not prioritized. Here, some controls are needed, for instance by asking to the participants to equally pay attention to both modalities and/or to evaluate the decoding in an auditory-alone condition and/or to use longer stimuli (i.e. more ecologic).

This point comes as a surprise to us for various reasons as we outline in the following. Most importantly, we believe that modelling the paradigm based on our previous work provides a critical strength, as we outline clearly in lines 69-79 in the manuscript. Participants were trained extensively in both the visual and auditory domain to eliminate biasing recollection effects, and the classification data analysis was designed to be insensitive to any such recollection effect.

The stimuli and novel experimental paradigm were carefully selected in order to resemble a real-world situation that requires a fast perceptual decision based on a burst of brief complementary perceptual evidence, as described in the introductory example (lines 33-34). The present design and stimuli should therefore be considered equally ecologic as the longer stimulus presentation suggested by the reviewer. Such an experimental design has specifically been overlooked in the multisensory literature, which is heavily focused on AV decisions using redundant stimuli, such as dots and sounds moving in the same direction or spatially co-localized audio-visual stimulus pairs. Contrary to the reviewer's assertion, we have not prioritised the visual categories over the auditory information and we have specifically and repeatedly instructed participants to pay attention to both modalities (exactly as the reviewer suggested). Interleaving V and AV trials ensured that participants could not predict the upcoming trial and they were therefore prepared to deal with both streams of information. We now make this point even clearer in the Methods section on lines 641-647.

Following on from this, as the reviewer pointed out, the selected sounds can be considered perceptually difficult and necessitating a training session, which we have indeed provided to our participants. As a matter of fact, participants received an equal amount of training trials on the visual and auditory stimuli. Thus, no additional memory component was added specifically to the AV condition of the task. Should somehow such recall effects exist in the AV condition, they should apply equally to both face and car trials. As we highlighted in our response to the previous comment and contrary to the reviewer's assertion, our multivariate discriminant analysis was specifically designed to separate out evidence

supporting a face-vs-car decision, and therefore any added recall effects present on both categories would not have contributed to the discrimination.

Lastly, modelling the present study on previous work (which we have replicated on several occasions over the last decade [Blank, Biele, Heekeren, & Philiastides, 2013; Diaz, Queirazza, & Philastides, 2017; Lou, Li, Philiastides, & Sajda, 2014; Philiastides & Sajda, 2006]) allowed us to validate the observed EEG components. As we note in the manuscript, the classifier topographies and latencies in the visual condition match those we reported previously, further confirming that the present data are not biased by some sort of participant-specific idiosyncrasies. Importantly, this also allowed us to dissociate any difference in neural evidence related to the task at hand that was due to the additional auditory evidence on AV trials. This increased amount of neural evidence was a result of the combination of information from both sensory channels (i.e., V and AV) as it surpassed the amount of evidence on visual trials alone (Figures 3d and 4d in the manuscript and the new figure under the first comment above).

In the training session, the participants had to learn to associate the very short auditory stimuli with one of the two categories. Actually, to perform the training session, learning to match sounds to only one category is sufficient. Like learning to recognize the human speech sounds only. Even normalized and embedded in some noise, human speech sounds are more homogeneous in their spectral contents than the car/street related sounds (that were squeaking tires or slammed door sounds). This disparity between categories may provide some explanation to the difference observed in the accuracy as well as in the responses time distribution, that were paralleled by the results of the nHDDM (i.e. starting point bias) . In link with this point, it would be important to have access to the associated statistical comparisons (i.e. difference in accuracy and RT between categories as well as conditions).

We were puzzled by the reviewer's argument (and subsequent logic) regarding differences in acoustic features as a trivial explanation of our data since they are neither compatible with our experimental procedures nor our reported results.

As pointed out in an earlier comment above, to avoid introducing a task-specific bias by training only one modality or one stimulus category, our experimental design included an equally long training procedure for visual trials, balanced across stimulus categories, on the same training day.

According to the reviewer's proposed argument, spectral differences between sound categories would result in easier decisions on AV trials, if mainly/only the sound is being used. It would follow that this category difference would drive the behavioural benefit reflected in improved decision accuracy on AV trials. In fact, spectrally different sound categories should permit rapid categorization of stimuli all the more – at least after practice. Thereby, the task would become easier which should translate into much shorter response times for AV trials. However, the reviewer's argument is not supported by the slower RTs we observed in the present study, and their explanation provided by our nHDDM.

Following on from this, the small starting point bias identified by the nHDDM model was comparable across sensory modalities, highlighting that any category-specific differences (face vs car) were not specific to one of the sensory modalities (either AV or V). As such, any potential idiosyncrasies in behaviour across stimulus categories were similarly represented across sensory modalities and were therefore removed when computing AV-V differences. We had made this point clear in the previous revision (lines 387-389 of the latest revised manuscript). We had also already reported formal statistics on the starting point bias (line 388) as well as descriptive statistics on behaviour associated with the

comparison of the stimulus categories (i.e., faces and cars; lines 392-393). As requested, we now also provide formal statistics of these comparisons in lines 394-396.

3) Some of the findings ask for further examination to assess how they may have influenced the main results and/or to clarify their meaning.

For instance, a latency analysis performed on the Early component demonstrated the existence of two groups in the participants. Since this is an unexpected result, it calls for additional analysis and should be incorporated as a variable of interest in the analyses performed. As an example, how many participants belong to each group? Does this two groups correspond to the participants for which the Late component amplitude is higher/lower in the AV condition as to compared to the V condition?

What the reviewer calls an unexpected result is in our view the natural heterogeneity of human performance and brain activity seen in any cognitive neuroimaging study. For the present data, this intersubject variability gives rise to two peaks in the stimulus-locked group-level classifier output (Figure 3a). As a result, peak times of the Early component vary between participants ($SD_V = 53.84\text{ms}$, $SD_{AV} = 57.52\text{ms}$).

To directly address this comment, we performed additional analyses to investigate whether these inter-individual differences in the timing of the Early (sensory) component may have affected the amplitude and timing of the Later (post-sensory) component. We performed these analyses across all participants (i.e., treating participants as a continuous variable) because in our view it does not make sense to divide participants into two sub-groups simply based on intersubject variability using an arbitrary cut-off.

Firstly, we tested the extent to which the individual onset times in the Early component predicted the difference in the Late component discriminator amplitudes across the two modalities (AV vs V). In other words, we tested whether “early encoders” who may have been more efficient in encoding the sensory evidence, caused the bulk of the differences that we observed across AV/V trials in late post-sensory processing. Specifically, we used a robust bend correlation between the subject-specific average peak onset time of the Early component across both sensory modality conditions and the difference in stimulus-locked Late component amplitude (AV-V) across participants. We found that Early component peak time is not significantly correlated with the amplitude of the Late component ($r_{\text{bend}} = -0.28$, $p = .071$). This result shows that the latency of the Early component had no leverage on the differences we observed in the Late component across modalities.

Secondly, we also examined the potential relationship between the subject-specific average peak onset time of the Early component across both sensory modality conditions and the peak time of the Late component. Once again, this correlation analysis indicated no significant relationship ($r_{\text{bend}} = .22$, $p = .1752$), thereby demonstrating that the latency of the Early encoding of the stimulus evidence did not affect the latency of participants’ post-sensory decision evidence. We added these correlation results and a brief description of the employed method to the results and method sections respectively (lines 247-253 and 881-884, respectively)

Taken together, these results offer further validation for the paper’s main conclusion that it is the strength of the post-sensory, rather than the early sensory evidence that is being amplified during rapid audiovisual decision-making. We now report these results in the revised manuscript on pages 10-12.

Another illustration relates to the drift rate difference between the Early and the Late component (Figure 5). It is interesting to see that the drift rate is larger than zero (Result section line 342). But even more interesting to note that for the Early component the parameter is between 0-1, and therefore reduces the impact of the Early component on the drift rates; which is not the case for the late component with a parameter above 1 in average.

In our view, the reviewer's observation basically validates our conclusion that the Late component amplitudes are a better index of the drift rate than the Early component amplitudes. Although both components contribute to the drift rate (regression coefficients are higher than 0 for both), the Late component trial-by-trial amplitudes correlate significantly better with the amount of decision evidence (represented by the drift rate in the model, lines 355-360 in the revised manuscript).

In link with this point, since the authors linked the Early component to sensory encoding (see lines 317, 339-344), it would make more sense to inform the non-decisional time (rather than the drift rate) with the amplitude of the early component.

As we have shown in previous studies (e.g., Philiastides, Heekeren, & Sajda, 2014; Ratcliff, Philiastides, & Sajda, 2009), the Early and Late components represent the amount of sensory and decision evidence respectively and their amplitudes correlate with the single-trial drift rates in the model (with a stronger contribution from the Late component). Given the timing of the Early component, it is indeed worth considering whether it also relates to the duration of sensory processing mechanisms captured in the HDDM by non-decision times.

Based on this rationale, we had already considered such an analysis (though not reported for brevity) in the previous round: the fits of the suggested model that scales non-decision time with the amplitude of the Early component were worse than the ones of the model we selected and reported in the revised manuscript (DIC = 1277 vs 517 for the chosen nHDDM). The suggested model also showed no relationship between non-decision times and the Early component γ 's (regression coefficients were not significantly different from 0 for both V and AV). Hence, we can safely say that the models considered in the current manuscript provide a more parsimonious account of the data. In fact, we view this finding as further evidence that increases in the non-decision times observed in AV trials are likely driven, specifically, by increased early encoding times of the added auditory information. We now present this analysis in the manuscript (lines 407-414).

Last, the authors began to envisage some of the issues I raised in the previous review, but their rapid analysis is only superficial (e.g. they evaluated if the low performers in the visual condition improved more in the audiovisual condition). To pursue in that direction, it would be pertinent to take into account coherence levels to assess if the participants tend to use more the auditory information when the image is noisier.

The analysis the reviewer requested had already been presented in the last round of reviews. The single-participant data displayed by visual coherence level in Figures 2b and 2e of the manuscript demonstrate that we neither observed a modality difference in accuracy (2b) nor response time (2e) that differed between visual coherence levels; as implied by the reviewer. Hence, mean/median measures of central tendency of the behavioural measures did not differ systematically by visual coherence level. Formally, during the last round of reviews, we provided strong evidence from a Bayesian modelling analysis against interaction effects between sensory modality and the level of visual evidence for both accuracy and response time data (please see lines 119 and 124 of the revised manuscript). In other words, since the distributions in Figures 2b and 2e are virtually similar across visual coherence levels, they illustrate our conjecture that participants did not benefit more from additional auditory evidence with decreasing amount of visual evidence.

These results are merely complemented by Figure 2c, which we added in the last round of reviews, demonstrating that (1) the majority of individual participants improved in the AV condition across all levels of visual evidence (coherence) and that (2) those participants who performed well in the V condition also performed well in the AV condition.

4) In the introduction, the authors undermine the current knowledge of the field. For instance in the fourth paragraph (lines 59 to 70), all the cited neuroimaging studies were published before 2013; even though some recent publications combined modeling and MEEG data (for instance, Rohe et al 2019 or Cao et al. 2019, cited later in the manuscript and published in Nat. Comms). Also, it would be informative for the community to have some links with more classical approaches. Here, one may think about the ERP community where some components have been linked to specific steps of decision making process (see for example the seminal work of O'Connell and colleagues).

We thank the reviewer for this suggestion and have addressed it by adding more citations to traditional ERP approaches among other literature specifically to the introduction. Some of which had already been cited in the previous introduction. (Angelaki et al., 2009; Bizley et al., 2016; Cao et al. 2019; Murray et al., 2004; Mercier et al., 2013; O'Connell et al., 2012; O'Connell et al., 2018; Rohe et al., 2019; Stein & Stanford, 2008).

These citations can now be found as citation numbers: 1, 2, 4, 10, 11, 18, 23, 31, 35.

Minors comments:

Abstract

Line 17: "'early' sensory processing benefits" can the authors clarify what they mean by early benefits?

To clarify the terminology for everyone, we amended the sentence in question in the abstract as follows: *'[...] explained by sensory (i.e., 'early') processing benefits or post-sensory (i.e., 'late') changes in decision dynamics'*.

Line 25: "multisensory behavioural improvements" The plural form is not correct here as only accuracy was improved, not RT.

The phrasing of this sentence was adapted to reflect the singular form of the improvement in decision accuracy.

'Using a neurally-informed drift diffusion model we demonstrate that a multisensory behavioural improvement in accuracy arises from an enhanced quality of the relevant decision evidence,[...]

Line 27: "high-level conceptualization" this is vague term, it should be clarified.

To clarify the term pointed out by the reviewer, we adopted the following alternate phrasing: *'[...] consistent with the emergence of multisensory evidence in higher-order brain areas'*

Materials and Methods

Lines 594-5: A sentence has been truncated.

Thank you for spotting this issue. We have amended the sentence now reading as follows:

'Stimuli presentation duration for all stimuli and tasks was set to 50ms for comparability between the training and testing days.' (pg. 30, lines 657–658)

Lines 707-709: This part of the sentence is confusing 'visual evidence in each modality'. The word 'condition' would suit better than the word 'modality'.

Since our experimental design included different conditions in both the modality dimension and the amount of visual evidence, we argue that the word 'condition' on its own bears the potential to be confusing as well. Therefore, we believe that adding the word 'condition' to the term 'in each modality', which now effectively reads *'in each sensory modality condition'* resolves potential confusion (line 767).

line 872: Does the negative sign means that the range for the late component is computed relative to the response?

Indeed, the negative sign indicates that these times are relative to the response. Since the last revision we have used the Late stimulus-locked component amplitudes for the modelling as stated in lines 332-336 of the results section (for consistency and direct comparisons with the Early stimulus-locked component). Line 872 was carried over from the original manuscript (referring to the original definition of our Late response-locked component) and should have been amended in the last revision (many thanks for spotting this!). We have now corrected the relevant sentence as follows:

"[...] the single-trial discriminator amplitudes of participant-specific stimulus-locked Early EEG components (individual peak A_z across V and AV) and stimulus-locked Late EEG components (individual peak A_z difference between AV and V in the time range established in Fig. 3b; 490 – 540ms [expanded further by 30ms on either side to account for the resulting A_z values being obtained with 60ms training windows centered on these times])." (pg. 42, lines 945-949)

line 875: Can the authors rationalized why the two components amplitude are not processed the same way since the later one is multiplied by C.

As reported in line 327 ff. of the previously revised manuscript, 'we tested three models where coherence scaled (a) γ_{Early} , (b) γ_{Late} , and (c) both components. We found the best fit for the model where coherence scaled γ_{Late} (DICs = 767, 517 and 661, respectively), indicating that the modulation of the Late component with the reliability of available evidence is predictive of the rate of evidence accumulation.' Hence, it follows that only the Late component amplitudes were modulated by the difficulty of the task (i.e., visual evidence represented by C in the equation previously reported on line 875), and thus they were better descriptors of the behavioural performance (accuracy) variations. In other words, the best fitting model indicates that the effect of task difficulty on behavioural performance is captured by post-sensory mechanisms (lines 343-347).

Results:

Line 160: To illustrate a correlation it is recommended to use square axis with identical scaling.

Despite the original depiction not obfuscating any results, as requested by the reviewer, we elongated the y-axis of Figure 2c in order for this figure to use a square axis in the revised version of the manuscript.

Line 196: For the early component, the mean peak times for the V and AV conditions appear to be exactly the same, down to the last decimal! This should be underline to ensure the reader that it is not a mistake or an oversight.

Indeed, we found the central tendency (i.e., mean) of the peak times of the Early component to be identical between sensory conditions (AV vs V) on the group level. Since this finding was intriguing, we double-checked the individual peak times, which varied as illustrated by their different standard deviations reported in the same line of the revised manuscript ($SD_V = 53.84\text{ms}$, $SD_{AV} = 57.52\text{ms}$; line

194). We argue that this variation demonstrates to the reader that this result arose from two different distributions with identical mean.

Line 331: This result is not surprising since the authors already show the link between stimulus phase coherence and the late component. This original finding should be reminded. As such this result only confirms the dissociation between the Early and Late component.

The reviewer is right in that this result is consistent with our previous work. We view this as a critical validation of previous work, specifically in the context of the observed multisensory effects, in order to provide concrete evidence linking the Late EEG component to post-sensory enhancements during multisensory decision making. We have now added additional content to make this point clear (lines 347-348).

Line 363: These results should be represented in a dedicated figure to show to the reader how parameters differs between categories per conditions (please include the variance, like using the mean absolute deviation).

The requested statistics regarding the starting point results including a variance indicator have been added to the revised manuscript (lines 375-376) to provide the reader with unobscured information on these data. Fig. 5e already displays these data as single-subject data for V and AV trials as well as their respective measure of central tendency.

Line 365: The corresponding values should be provided.

The requested average drift rates for V and AV car and face trials have been added to the latest revised version of the manuscript on lines 389-390 as follows:

'[...] and higher drift rate for car choices, in both the V and AV conditions ($\delta_{car-V} = -0.92 \pm 0.14$, $\delta_{face-V} = 0.15 \pm 0.13$, $\delta_{car-AV} = -0.96 \pm 0.14$, $\delta_{face-AV} = 0.07 \pm 0.12$ – positive [negative] signs indicate face [car] choices).'

Line 369: The bias in the starting point is rather the origin, not the outcome, of the accuracy difference.

This is a useful suggestion, indeed. We have amended the sentence as follows (lines 401-402):

'These results suggest that the bias in the starting point is likely driving the accuracy difference between face and car choices [...].'

Line 374: This statement is hurried; values and statistical tests must be provided.

We have added the statistical tests for the starting point and boundary separation differences between V and AV on lines 375-377. In the previous paragraph where we describe the modelling results: ($\beta_V = 0.5475 \pm 0.005$, $\beta_{AV} = 0.5495 \pm 0.005$; paired t-test: $t(39) = -0.4964$, $p = .6224$; Fig. 5e and $\alpha_V = 1.13 \pm 0.03$, $\alpha_{AV} = 0.12 \pm 0.03$; paired t-test: $t(39) = 0.8191$, $p = .4177$; Fig. 5f).

Discussion

Lines 455-462: This paragraph lacks of clarity, can the authors make their point clearer?

We have amended this paragraph to clarify our point that the improvements in drift rate on multisensory trials (i.e., faster integration of sensory evidence), which we have observed in the present study, converge with modelling results from a previous multisensory study employing a different task and different modalities (visual and vestibular; Drugowitsch et al., 2014). These converging modelling findings are all the more intriguing as they arose despite differences in the chosen modalities and overall behavioural outcomes, highlighting the domain generality of these reported effects. This revised paragraph can now be found in lines 514-523. of the revised manuscript.

Reviewer #3 (Remarks to the Author):

The authors have done an impressive job in addressing my concerns about the previous version of the manuscript.

Signed,
Sebastian Gluth

Reviewers' Comments:

Reviewer #3:

Remarks to the Author:

In my view, the authors have done a good job in finding a balance between addressing outstanding concerns raised by Reviewer #2 that were indeed important, and dismissing those points that were irrelevant (because they were based either on misunderstandings or on pure speculations).

Signed,

Sebastian Gluth